# Differentially-Private Federated Linear Bandits

**Abhimanyu Dubey** and **Alex Pentland**
Media Lab and Institute for Data, Systems and Society
Massachusetts Institute of Technology
{dubeya, pentland}@mit.edu

## Abstract

The rapid proliferation of decentralized learning systems mandates the need for differentially-private cooperative learning. In this paper, we study this in context of the contextual linear bandit: we consider a collection of agents cooperating to solve a common contextual bandit, while ensuring that their communication remains private. For this problem, we devise FEDUCB, a multiagent private algorithm for both centralized and decentralized (peer-to-peer) federated learning. We provide a rigorous technical analysis of its utility in terms of regret, improving several results in cooperative bandit learning, and provide rigorous privacy guarantees as well. Our algorithms provide competitive performance both in terms of pseudoregret bounds and empirical benchmark performance in various multi-agent settings.

## 1 Introduction

The multi-armed bandit is the classical sequential decision-making problem, involving an agent sequentially choosing actions to take in order to maximize a (stochastic) reward [44]. It embodies the central *exploration-exploitation* dilemma present in sequential decision-making. Practical applications of the multi-armed bandit range from recommender systems [52] and anomaly detection [11] to clinical trials [15] and finance [24]. Increasingly, however, such large-scale applications are becoming *decentralized*, as their data is often located with different entities, and involves cooperation between these entities to maximize performance [14, 22]. This paradigm is now known as *federated learning*[1].

The objective of the federated paradigm is to allow cooperative estimation with larger amounts of data (from multiple clients, devices, etc.) while keeping the data decentralized [27]. There has been a surge of interest in this problem from both academia and industry, owing to its overall applicability. Most research on provably private algorithms in the federated setting has been on distributed supervised learning [28] and optimization [20]. The contextual bandit problem, however, is a very interesting candidate for private methods, since the involved contexts and rewards both typically contain sensitive user information [38]. There is an increasing body of work on online learning and multi-armed bandits in cooperative settings [13, 31, 39], and private single-agent learning [41, 38], but methods for private *federated* bandit learning are still elusive, despite their immediate applicability.

**Contributions**. In this paper, we study the *federated* contextual bandit problem under constraints of differential privacy. We consider two popular paradigms: (a) *centralized* learning, where a central controller coordinates different clients [27, 49] and (b) *decentralized* peer-to-peer learning *with delays*, where agents communicate directly with each other, without a controller [39, 31, 30, 32]. We provide a rigorous formulation of $(\varepsilon, \delta)$-differential privacy in the *federated* contextual bandit, and present two variants of FEDUCB, the first federated algorithm ensuring that each agent is private with respect to the data from all other agents, and provides a parameter to control communication.

Next, we prove rigorous bounds on the cumulative group pseudoregret obtained by FEDUCB. In the centralized setting, we prove a high probability regret bound of $\widetilde{O}(d^{3/4}\sqrt{MT/\varepsilon})$ which matches the non-private bound in terms of its dependence on $T$ and $M$. In the decentralized case, we prove a corresponding regret bound of $\widetilde{O}(d^{3/4}\sqrt{(\text{diameter}(\mathcal{G}))MT/\varepsilon})$, where $\mathcal{G}$ is the communication network between agents. In addition to the regret bounds, we present a novel analysis of communication complexity, and its connections with the privacy budget and regret.

## 2    Related Work

**Multi-Agent and Distributed Bandits**. Bandit learning in multi-agent distributed settings has received attention from several academic communities. Channel selection in distributed radio networks consider the (context-free) multi-armed bandit with collisions [35, 37, 36] and cooperative estimation over a network with delays [31, 30, 32]. For the contextual case, recent work has considered non-private estimation in networks with delays [12, 13, 48, 29]. A closely-related problem is that of bandits with side information [8, 5], where the single learner obtains multiple observations every round, similar to the multi-agent communicative setting. Our work builds on the remarkable work of Abbasi-Yadkori *et al.*[1], which in turn improves the LinUCB algorithm introduced in [33].

**Differential Privacy**. Our work utilizes *differential privacy*, a cryptographically-secure privacy framework introduced by Dwork [16, 18] that requires the behavior of an algorithm to fluctuate only slightly (in probability) with any change in its inputs. A technique to maintain differential privacy for the continual release of statistics was introduced in [10, 17], known as the *tree-based* algorithm that privatizes the partial sums of $n$ entries by adding at most $\log n$ noisy terms. This method has been used to preserve privacy across several online learning problems, including convex optimization [26, 25], online data analysis [23], collaborative filtering [6] and data aggregation [9]. In the single-agent bandit setting, differential privacy using tree-based algorithms have been explored in the multi-armed case [43, 40, 46] and the contextual case [41]. In particular, our work uses several elements from [41], extending their single-agent results to the federated multi-agent setting. For the multi-agent multi-armed (i.e., context-free) bandit problem, differentially private algorithms have been devised for the centralized [45] and decentralized [14] settings. Empirically, the advantages of privacy-preserving contextual bandits has been demonstrated in the work of Malekzadeh *et al.*[38], and Hannun *et al.*[21] consider a centralized multi-agent contextual bandit algorithm that use secure multi-party computations to provide privacy guarantees (both works do not have any regret guarantees).

To the best of our knowledge, this paper is the first to investigate differential privacy for contextual bandits in the federated learning setting, in both centralized and decentralized environments. Our work is most closely related to the important work of [1, 41], extending it to federated learning.

## 3    Background and Preliminaries

We use boldface to represent vectors, e.g., $\boldsymbol{x}$, and matrices, e.g., $\boldsymbol{X}$. For any matrix $\boldsymbol{X}$, its Gram matrix is given by $\boldsymbol{X}^\top \boldsymbol{X}$. Any symmetric matrix $\boldsymbol{X}$ is positive semi-definite (PSD) if $\boldsymbol{y}^\top \boldsymbol{X}\boldsymbol{y} \geq 0$ for any vector $\boldsymbol{y}$, and we denote PSD by $\boldsymbol{X} \succcurlyeq \boldsymbol{0}$. For any PSD matrix $\boldsymbol{X}$ we denote the ellipsoid $\boldsymbol{X}$-norm of vector $\boldsymbol{y}$ as $\|\boldsymbol{y}\|_{\boldsymbol{X}} = \sqrt{\boldsymbol{y}^\top \boldsymbol{X}\boldsymbol{y}}$. For two matrices $\boldsymbol{X}$ and $\boldsymbol{Y}$, $\boldsymbol{X} \succcurlyeq \boldsymbol{Y}$ implies that $\boldsymbol{X} - \boldsymbol{Y} \succcurlyeq \boldsymbol{0}$. For any PSD matrix $\boldsymbol{X}$ and vectors $\boldsymbol{u}, \boldsymbol{v}$, we denote the $\boldsymbol{X}$-ellipsoid inner product as $\langle \boldsymbol{u}, \boldsymbol{v}\rangle_{\boldsymbol{X}} = \boldsymbol{u}^\top \boldsymbol{X}\boldsymbol{v}$, and drop the subscript when $\boldsymbol{X} = \boldsymbol{I}$. We denote the set $\{a, a+1, ..., b-1, b\}$ by the shorthand $[a, b]$, and if $a = 1$ we refer to it as $[b]$. We denote the $\gamma^{th}$ power of graph $\mathcal{G}$ as $\mathcal{G}_\gamma$ ($\mathcal{G}_\gamma$ has edge $(i, j)$ if the shortest distance between $i$ and $j$ in $\mathcal{G}$ is $\leq \gamma$). $N_\gamma(v)$ denotes the set of nodes in $\mathcal{G}$ at a distance of at most $\gamma$ (including itself) from node $v \in \mathcal{G}$.

**Federated Contextual Bandit**. This is an extension of the linear contextual bandit [33, 1] involving a set of $M$ agents. At every trial $t \in [T]$, each agent $i \in [M]$ is presented with a *decision set* $\mathcal{D}_{i,t} \subset \mathbb{R}^d$ from which it selects an action $\boldsymbol{x}_{i,t} \in \mathbb{R}^d$. It then obtains a reward $y_{i,t} = \boldsymbol{x}_{i,t}^\top \boldsymbol{\theta}^* + \eta_{i,t}$ where $\boldsymbol{\theta}^* \in \mathbb{R}^d$ is an unknown (but fixed) parameter and $\eta_{i,t}$ is a noise parameter sampled i.i.d. every trial for every agent. The objective of the agents is to minimize the cumulative *group pseudoregret*:[2]

$$\mathcal{R}_M(T) = \sum_{i=1,t=1}^{M,T} \langle \boldsymbol{x}_{i,t}^* - \boldsymbol{x}_{i,t}, \boldsymbol{\theta}^* \rangle, \text{ where } \boldsymbol{x}_{i,t}^* = \arg\max_{\boldsymbol{x} \in \mathcal{D}_{i,t}} \langle \boldsymbol{x}, \boldsymbol{\theta}^* \rangle \text{ is optimal.}$$

In the single-agent setting, the optimal regret is $\widetilde{O}(\sqrt{dT})$, which has been matched by a variety of algorithms, most popular of them being the ones based on upper confidence bounds (UCB) [1, 33, 4]. In *non-private* distributed contextual bandits, a group pseudoregret of $\widetilde{O}(\sqrt{dMT})$ has been achieved [48, 49], which is the order-optimality we seek in our algorithms as well.

*Assumptions.* We assume: *(a)* bounded action set: $\forall i, t, \|\boldsymbol{x}_{i,t}\| \leq L$, *(b)* bounded mean reward: $\forall \boldsymbol{x}, \langle \boldsymbol{\theta}^*, \boldsymbol{x} \rangle \leq 1$, *(c)* bounded target parameter: $\|\boldsymbol{\theta}^*\| \leq S$, *(d)* sub-Gaussian rewards: $\forall i, t, \eta_{i,t}$ is $\sigma$-sub-Gaussian, *(e)* bounded decision set: $\forall i, t \; \mathcal{D}_{i,t}$ is compact, *(f)* bounded reward: $\forall i, t, |y_{i,t}| \leq B$.[3]

**Differential Privacy**. The contextual bandit problem involves two sets of variables that any agent must private to the other participating agents – the available decision sets $(\mathcal{D}_{i,t})_{t \in [T]}$ and observed rewards $(y_{i,t})_{t \in [T]}$. The adversary model assumed here is to prevent any two colluding agents $j$ and $k$ to obtain non-private information about any specific element in agent $i$'s history. That is, we assume that each agent is a trusted entity that interacts with a new user at each instant $t$. Therefore, the context set $(\mathcal{D}_{i,t})$ and outcome $(y_{i,t})$ are sensitive variables that the user trusts only with the agent $i$. Hence, we wish to keep $(\mathcal{D}_{i,t})_{t \in [T]}$ private. However, the agent only stores the chosen actions $(\boldsymbol{x}_{i,t})_{t \in [T]}$ (and not all of $\mathcal{D}_{i,t}$), and hence making our technique differentially private with respect to $((\boldsymbol{x}_{i,t}, y_{i,t}))_{t \in [T]}$ will suffice. We first denote two sequences $S_i = ((\boldsymbol{x}_{i,t}, y_{i,t}))_{t \in [T]}$ and $S_i' = ((\boldsymbol{x}_{i,t}', y_{i,t}'))_{t \in [T]}$ as $t-$neighbors if for each $t' \neq t$, $(\boldsymbol{x}_{i,t}, y_{i,t}) = (\boldsymbol{x}_{i,t}', y_{i,t}')$. We can now provide the formal definition for federated differential privacy:

**Definition 1** (Federated Differential Privacy). *In a federated learning setting with $M \geq 2$ agents, a randomized multiagent contextual bandit algorithm $A = (A_i)_{i=1}^M$ is $(\varepsilon, \delta, M)$-federated differentially private under continual multi-agent observation if for any $i, j$ s.t. $i \neq j$, any $t$ and set of sequences $\boldsymbol{S}_i = (S_k)_{k=1}^M$ and $\boldsymbol{S}_i' = (S_k)_{k=1, k \neq i}^M \cup S_i'$ such that $S_i$ and $S_i'$ are t-neighboring, and any subset of actions $\mathcal{S}_j \subset \mathcal{D}_{j,1} \times \mathcal{D}_{j,2} \times ... \times \mathcal{D}_{j,T}$ of actions, it holds that:*

$$\mathbb{P}\left(A_j\left(\boldsymbol{S}_i\right) \in \mathcal{S}_j\right) \leq e^\varepsilon \cdot \mathbb{P}\left(A_j\left(\boldsymbol{S}_i'\right) \in \mathcal{S}_j\right) + \delta.$$

Our notion of federated differential privacy is formalizing the standard intuition that "the action chosen by any agent must be sufficiently impervious (in probability) to any single $(\boldsymbol{x}, y)$ pair from any other agent". Here, we essentially lift the definition of joint differential privacy [41] from the individual $(\boldsymbol{x}, y)$ level to the entire history $(\boldsymbol{x}_t, y_t)_t$ for each participating agent. Note that our definition in its current form does not require each algorithm to be private with respect to its own history, but only the histories belonging to other agents, i.e., each agent can be trusted with its own data. This setting can also be understood as requiring all *outgoing communication* from any agent to be *locally differentially private* [51] to the personal history $(\boldsymbol{x}_t, y_t)_t$. We can alternatively relax this assumption and assume that the agent cannot be trusted with its own history, in which case the notion of joint or local DP at the individual level (i.e., $(\boldsymbol{x}_t, y_t)$ ) must be considered, as done in [51, 41].

The same guarantee can be obtained if each agent $A_i$ is $(\varepsilon, \delta)$-differentially private (in the standard contextual bandit sense, see [41]) with respect to any other agent $j$'s observations, for all $j$. A composition argument [16][4] over all $M$ agents would therefore provide us $(\sqrt{2M \log(1/\delta')}\varepsilon + M\varepsilon(e^\varepsilon - 1), M\delta + \delta')$-differential privacy with respect to the overall sequence. To keep the notation simple, however, we adopt the $(\varepsilon, \delta, M)$ format.

## 4 Federated LinUCB with Differential Privacy

In this section, we introduce our algorithm for federated learning with differential privacy. For the remainder of this section, for exposition, we consider the single-agent setting and drop an additional index subscript we use in the actual algorithm (e.g., we refer to the action at time $t$ as $\boldsymbol{x}_t$ and not $\boldsymbol{x}_{i,t}$ for agent $i$). We build on the celebrated LinUCB algorithm, an application of the optimism heuristic to the linear bandit case [33, 1], designed for the single-agent problem. The central idea of the algorithm is, at every round $t$, to construct a *confidence set* $\mathcal{E}_t$ that contains $\boldsymbol{\theta}^*$ with high probability, followed by computing an upper confidence bound on the reward of

**Algorithm 1** CENTRALIZED FEDUCB$(D, M, T, \rho_{\min}, \rho_{\max})$

1: **Initialization**: $\forall i$, set $\boldsymbol{S}_{i,1} \leftarrow M\rho_{\min}\boldsymbol{I}, \boldsymbol{s}_{i,1} \leftarrow \mathbf{0}, \widehat{\boldsymbol{Q}}_{i,0} \leftarrow \mathbf{0}, \boldsymbol{U}_{i,1} \leftarrow \mathbf{0}, \bar{\boldsymbol{u}}_{i,1} \leftarrow \mathbf{0}$.
2: **for** For each iteration $t \in [T]$ **do**
3:    **for** For each agent $i \in [M]$ **do**
4:       Set $\boldsymbol{V}_{i,t} \leftarrow \boldsymbol{S}_{i,t} + \boldsymbol{U}_{i,t}, \tilde{\boldsymbol{u}}_{i,t} \leftarrow \boldsymbol{s}_{i,t} + \bar{\boldsymbol{u}}_{i,t}$.
5:       Receive $\mathcal{D}_{i,t}$ from environment.
6:       Compute regressor $\bar{\boldsymbol{\theta}}_{i,t} \leftarrow \boldsymbol{V}_{i,t}^{-1}\tilde{\boldsymbol{u}}_{i,t}$.
7:       Compute $\beta_{i,t}$ following Proposition 2.
8:       Select $\boldsymbol{x}_{i,t} \leftarrow \arg\max_{\boldsymbol{x} \in \mathcal{D}_{i,t}} \langle \boldsymbol{x}, \bar{\boldsymbol{\theta}}_{i,t} \rangle + \beta_{i,t} \|\boldsymbol{x}\|_{\boldsymbol{V}_{i,t}^{-1}}$.
9:       Obtain $y_{i,t}$ from environment.
10:      Update $\boldsymbol{U}_{i,t+1} \leftarrow \boldsymbol{U}_{i,t} + \boldsymbol{x}_{i,t}\boldsymbol{x}_{i,t}^{\top}, \boldsymbol{u}_{i,t+1} \leftarrow \boldsymbol{u}_{i,t} + \boldsymbol{x}_{i,t}y_{i,t}$.
11:      Update $\widehat{\boldsymbol{Q}}_{i,t} \leftarrow \widehat{\boldsymbol{Q}}_{i,t-1} + [\boldsymbol{x}_{i,t}^{\top} \ y_{i,t}]^{\top}[\boldsymbol{x}_{i,t}^{\top} \ y_{i,t}]$
12:      **if** $\log\det\left(\boldsymbol{V}_{i,t} + \boldsymbol{x}_{i,t}\boldsymbol{x}_{i,t}^{\top} + M(\rho_{\max} - \rho_{\min})\boldsymbol{I}\right) - \log\det\left(\boldsymbol{S}_{i,t}\right) \geq {}^{D}\!/\!_{\Delta t_i}$ **then**
13:        SYNCHRONIZE $\leftarrow$ TRUE.
14:      **end if**
15:      **if** SYNCHRONIZE **then**
16:        [$\forall$ AGENTS] Agent sends $\widehat{\boldsymbol{Q}}_{i,t} \rightarrow$ PRIVATIZER and gets $\widehat{\boldsymbol{U}}_{i,t+1}, \widehat{\boldsymbol{u}}_{i,t+1} \leftarrow$ PRIVATIZER.
17:        [$\forall$ AGENTS] Agent communicates $\widehat{\boldsymbol{U}}_{i,t+1}, \widehat{\boldsymbol{u}}_{i,t+1}$ to controller.
18:        [CONTROLLER] Compute $\boldsymbol{S}_{t+1} \leftarrow \sum_{i=1}^{M} \widehat{\boldsymbol{U}}_{i,t+1}, \boldsymbol{s}_{t+1} \leftarrow \sum_{i=1}^{M} \widehat{\boldsymbol{u}}_{i,t+1}$.
19:        [CONTROLLER] Communicate $\boldsymbol{S}_{t+1}, \boldsymbol{s}_{i,t+1}$ back to agent.
20:        [$\forall$ AGENTS] $\boldsymbol{S}_{i,t+1} \leftarrow \boldsymbol{S}_{t+1}, \boldsymbol{s}_{i,t+1} \leftarrow \boldsymbol{s}_{t+1}$.
21:        [$\forall$ AGENTS] $\widehat{\boldsymbol{Q}}_{i,t+1} \leftarrow \mathbf{0}$.
22:      **else**
23:        $\boldsymbol{S}_{i,t+1} \leftarrow \boldsymbol{S}_{i,t}, \boldsymbol{s}_{i,t+1} \leftarrow \boldsymbol{s}_{i,t}, \Delta t_i \leftarrow \Delta t_i + 1$.
24:        $\Delta t_i \leftarrow 0, \boldsymbol{U}_{i,t+1} \leftarrow \mathbf{0}, \bar{\boldsymbol{u}}_{i,t+1} \leftarrow \mathbf{0}$.
25:      **end if**
26:    **end for**
27: **end for**

each action within the decision set $\mathcal{D}_t$, and finally selecting the action with the largest UCB, i.e., $\boldsymbol{x}_t = \arg\max_{\boldsymbol{x} \in \mathcal{D}_t} (\max_{\boldsymbol{\theta} \in \mathcal{E}_t} \langle \boldsymbol{x}, \boldsymbol{\theta} \rangle)$. The confidence set is an ellipsoid centered on the regularized linear regression estimate (for $\boldsymbol{X}_{<t} = \left[\boldsymbol{x}_1^{\top} \ \boldsymbol{x}_2^{\top} \ ... \ \boldsymbol{x}_{t-1}^{\top}\right]^{\top}$ and $\boldsymbol{y}_{<t} = [y_1 \ y_2 \ ... \ y_{t-1}]^{\top}$):

$$\mathcal{E}_t := \left\{\boldsymbol{\theta} \in \mathbb{R}^d : \|\boldsymbol{\theta} - \hat{\boldsymbol{\theta}}_t\|_{\boldsymbol{V}_t} \leq \beta_t\right\}, \text{ where } \hat{\boldsymbol{\theta}}_t := \arg\min_{\boldsymbol{\theta} \in \mathbb{R}^d} \left[\|\boldsymbol{X}_{<t}\boldsymbol{\theta} - \boldsymbol{y}_{<t}\|_2^2 + \|\boldsymbol{\theta}\|_{\boldsymbol{H}_t}^2\right].$$

The regression solution can be given by $\hat{\boldsymbol{\theta}}_t := (\boldsymbol{G}_t + \boldsymbol{H}_t)^{-1}\boldsymbol{X}_{<t}^{\top}\boldsymbol{y}_{<t}$, where $\boldsymbol{G}_t = \boldsymbol{X}_{<t}^{\top}\boldsymbol{X}_{<t}$ is the Gram matrix of actions, $\boldsymbol{H}_t$ is a (time-varying) regularizer, and $\beta_t$ is an appropriately chosen exploration parameter. Typically in non-private settings, the regularizer is constant, i.e., $\boldsymbol{H}_t = \lambda\boldsymbol{I} \ \forall t, \lambda > 0$ [1, 33], however, in our case, we will carefully select $\boldsymbol{H}_t$ to introduce privacy, using a strategy similar to [41]. Given $\boldsymbol{V}_t = \boldsymbol{G}_t + \boldsymbol{H}_t$, let $\text{UCB}_t(\boldsymbol{x}; \boldsymbol{\theta}) = \langle \boldsymbol{\theta}, \boldsymbol{x} \rangle + \beta_t\|\boldsymbol{x}\|_{\boldsymbol{V}_t^{-1}}$.

In the federated setting, since there are $M$ learners that have distinct actions, the communication protocol is a key component of algorithm design: communication often creates *heterogeneity* between agents, e.g., for any two agents, their estimators $(\hat{\boldsymbol{\theta}}_t)$ at any instant are certainly distinct, and the algorithm must provide a control over this heterogeneity, to bound the group regret. We additionally require that communication between agents is $(\varepsilon, \delta)$-private, making the problem more challenging.

## 4.1 Centralized Environment with a Controller

We first consider the centralized communciation environment where there exists a controller that coordinates communication between different agents, as is typical in large-scale distributed learning. We consider a set of $M$ agents that each are interacting with the contextual bandit, and periodically communicate with the controller, that synchronizes them with other agents. We present the algorithm CENTRALIZED FEDUCB in Algorithm 1 that details our approach.

Algorithm 1 works as follows. Consider an agent $i$, and assume that synchronization had last taken place at instant $t'$. At any instant $t > t'$, the agent has two sets of parameters - **(A)** the first being all observations up to instant $t'$ for all $M$ agents (including itself) and **(B)** the second being its own observations from instant $t'$ to $t$. Since **(A)** includes samples from other agents, these

are privatized, and represented as the Gram matrix $\boldsymbol{S}_{t'+1} = \sum_{i \in [M]} \widehat{\boldsymbol{U}}_{i,t'+1}$ and reward vector $\boldsymbol{s}_{t'+1} = \sum_{i \in [M]} \widehat{\boldsymbol{u}}_{i,t'+1}$. Algorithm 1 privatizes its own observations as well (for simplicity in analysis) and hence $\boldsymbol{S}, \boldsymbol{s}$ are identical for all agents at all times. Moreover, since the group parameters are noisy variants of the original parameters, i.e., $\widehat{\boldsymbol{U}}_{i,t} = \boldsymbol{G}_{i,t} + \boldsymbol{H}_{i,t}$ and $\widehat{\boldsymbol{u}}_{i,t} = \boldsymbol{u}_{i,t} + \boldsymbol{h}_{i,t}$ (where $\boldsymbol{H}_{i,t}$ and $\boldsymbol{h}_{i,t}$ are perturbations), we can rewrite $\boldsymbol{S}_t, \boldsymbol{s}_t$ as (for any instant $t > t'$),

$$\boldsymbol{S}_t = \sum_{i \in [M]} \left( \sum_{\tau=1}^{t'} \boldsymbol{x}_{i,\tau} \boldsymbol{x}_{i,\tau}^\top + \boldsymbol{H}_{i,t'} \right), \boldsymbol{s}_t = \sum_{i \in [M]} \left( \sum_{\tau=1}^{t'} y_{i,\tau} \boldsymbol{x}_{i,\tau} + \boldsymbol{h}_{i,t'} \right). \tag{1}$$

When we combine the group parameters with the local (unsynchronized) parameters, we obtain the final form of the parameters for any agent $i$ as follows (for any instant $t > t'$):

$$\boldsymbol{V}_{i,t} = \sum_{\tau=t'}^{t-1} \boldsymbol{x}_{i,\tau} \boldsymbol{x}_{i,\tau}^\top + \boldsymbol{S}_t, \tilde{\boldsymbol{u}}_{i,t} = \sum_{\tau=t'}^{t-1} y_{i,\tau} \boldsymbol{x}_{i,\tau} + \boldsymbol{s}_t \tag{2}$$

Then, with a suitable sequence $(\beta_{i,t})_t$, the agent selects the action following the linear UCB objective:

$$\boldsymbol{x}_{i,t} = \underset{\boldsymbol{x} \in \mathcal{D}_{i,t}}{\arg\max} \left( \langle \widehat{\boldsymbol{\theta}}_{i,t}, \boldsymbol{x} \rangle + \beta_{i,t} \|\boldsymbol{x}\|_{\boldsymbol{V}_{i,t}^{-1}} \right) \text{ where } \widehat{\boldsymbol{\theta}}_{i,t} = \boldsymbol{V}_{i,t}^{-1} \tilde{\boldsymbol{u}}_{i,t}. \tag{3}$$

The central idea of the algorithm is to therefore carefully perturb the Gram matrices $\boldsymbol{V}_{i,t}$ and the reward vector $\boldsymbol{u}_{i,t}$ with random noise $(\boldsymbol{H}_{i,t}, \boldsymbol{h}_{i,t})$ based on the sensitivity of these elements and the level of privacy required, as is typical in the literature [41]. First, each agent updates its local (unsynchronized) estimates. These are used to construct the UCB in a manner identical to the standard OFUL algorithm [1]. If, for any agent $i$, the log-determinant of the local Gram matrix exceeds the synchronized Gram matrix ($\boldsymbol{S}_{i,t}$) by an amount $D/\Delta t_i$ (where $\Delta t_i$ is the time since the last synchronization), then it sends a signal to the controller, that synchronizes *all* agents with their latest action/reward pairs. The synchronization is done using a *privatized* version of the Gram matrix and rewards, carried out by the subroutine PRIVATIZER (Section 4.3, Alg. 2). This synchronization ensures that the *heterogeneity* between the agents is controlled, allowing us to control the overall regret and limit communication as well:

**Proposition 1** (Communication Complexity)**.** *If Algorithm 1 is run with threshold D, then total rounds of communication $n$ is upper bounded by* $2\sqrt{(dT/D)} \cdot \log\left(\rho_{\max}/\rho_{\min} + TL^2/d\rho_{\min}\right) + 4$.

Now, the perturbations $\boldsymbol{H}_{i,t}, \boldsymbol{h}_{i,t}$ are designed keeping the privacy setting in mind. In our paper, we defer this to the subsequent section in a subroutine known as PRIVATIZER, and concentrate on the performance guarantees first. The PRIVATIZER subroutine provides suitable perturbations based on the privacy budget ($\varepsilon$ and $\delta$), and is inspired by the single-agent private algorithm of [41]. In this paper, we assume these budgets to be identical for all agents (however, the algorithm and analysis hold for unique privacy budgets as well, as long as a lower bound on the budgets is known). In turn, the quantities $\varepsilon$ and $\delta$ affect the algorithm (and regret) via the quantities $\rho_{\min}, \rho_{\max}$, and $\kappa$ which can be understood as spectral bounds on $\boldsymbol{H}_{i,t}, \boldsymbol{h}_{i,t}$.

**Definition 2** (Sparsely-accurate $\rho_{\min}, \rho_{\max}$ and $\kappa$)**.** *Consider a subsequence $\bar{\boldsymbol{\sigma}}$ of $[T] = 1, ..., T$ of size $n$. The bounds $0 \le \rho_{\min} \le \rho_{\max}$ and $\kappa$ are $(\alpha/2nM, \bar{\boldsymbol{\sigma}})$-accurate for $(\boldsymbol{H}_{i,t})_{i \in [M], t \in \bar{\boldsymbol{\sigma}}}$ and $(\boldsymbol{h}_{i,t})_{i \in [M], t \in \bar{\boldsymbol{\sigma}}}$, if, for each round $t \in \bar{\boldsymbol{\sigma}}$ and agent $i$:*

$$\|\boldsymbol{H}_{i,t}\| \le \rho_{\max}, \ \left\|\boldsymbol{H}_{i,t}^{-1}\right\| \le 1/\rho_{\min}, \ \|\boldsymbol{h}_{i,t}\|_{\boldsymbol{H}_{i,t}^{-1}} \le \kappa; \text{ with probability at least } (1 - \alpha/2nM).$$

The motivation for obtaining accurate bounds $\rho_{\min}, \rho_{\max}$ and $\kappa$ stems from the fact that in the non-private case, the quantities that determine regret are not stochastic conditioned on the obtained sequence $(\boldsymbol{x}_t, y_t)_{t \in [T]}$, whereas the addition of stochastic regularizers in the private case requires us to have control over their spectra to achieve any meaningful regret. To form the UCB, recall that we additionally require a suitable exploration sequence $\beta_{i,t}$ for each agent, which is defined as follows.

**Definition 3** (Accurate $(\beta_{i,t})_{i \in [M], t \in [T]}$)**.** *A sequence $(\beta_{i,t})_{i \in [M], t \in [T]}$ is $(\alpha, M, T)$-accurate for $(\boldsymbol{H}_{i,t})_{i \in [M], t \in [T]}$ and $(\boldsymbol{h}_{i,t})_{i \in [M], t \in [T]}$, if it satisfies $\|\tilde{\boldsymbol{\theta}}_{i,t} - \boldsymbol{\theta}^*\|_{\boldsymbol{V}_{i,t}} \le \beta_{i,t}$ with probability at least $1 - \alpha$ for all rounds $t = 1, ..., T$ and agents $i = 1, ..., M$ simultaneously.*

**Proposition 2.** *Consider an instance of the problem where synchronization occurs exactly $n$ times on instances $\bar{\boldsymbol{\sigma}}$, up to and including $T$ trials, and $\rho_{\min}, \rho_{\max}$ and $\kappa$ are $(\alpha/2nM)$-accurate. Then, for Algorithm 1, the sequence $(\beta_{i,t})_{i\in[M],t\in[T]}$ is $(\alpha, M, T)$-accurate where:*

$$\beta_{i,t} := \sigma\sqrt{2\log(^2/\alpha) + d\log\left(\det(\boldsymbol{V}_{i,t})\right) - d\log(M\rho_{\min})} + S\sqrt{M\rho_{\max}} + \kappa\sqrt{M}.$$

The key point here is that for the desired levels of privacy $(\varepsilon, \delta)$ and synchronization rounds $(n)$, we can calculate appropriate $\rho_{\min}, \rho_{\max}$ and $\kappa$, which in turn provide us a UCB algorithm with guarantees. Our basic techniques are similar to [1, 41], however, the multi-agent setting introduces several new aspects in the analysis, such as the control of heterogeneity.

**Theorem 1** (Private Group Regret with Centralized Controller). *Assuming Proposition 2 holds, and synchronization occurs in at least $n = \Omega\left(d\log\left(\rho_{\max}/\rho_{\min} + TL^2/d\rho_{\min}\right)\right)$ rounds, Algorithm 1 obtains the following group pseudoregret with probability at least $1 - \alpha$:*

$$\mathcal{R}_M(T) = O\left(\sigma\sqrt{MTd}\left(\log\left(\rho_{\max}/\rho_{\min} + TL^2/d\rho_{\min}\right) + \sqrt{\log^2/\alpha} + S\sqrt{M\rho_{\max}} + \kappa\sqrt{M}\right)\right).$$

This regret bound is obtained by setting $D = 2Td\left(\log\left(\rho_{\max}/\rho_{\min} + TL^2/d\rho_{\min}\right) + 1\right)^{-1}$, which therefore ensures $O(M\log T)$ total communication (by Proposition 1). However, the remarks next clarify a more sophisticated relationship between communication, privacy and regret:

**Remark 1** (Communication Complexity and Regret). *Theorem 1 assumes that communication is essentially $O(M\log T)$. This rate (and corresponding $D$) is chosen to provide a balance between privacy and utility, and in fact, can be altered depending on the application. In the Appendix, we demonstrate that when we allow $O(MT)$ communication (i.e., synchronize every round), the regret can be improved by a factor of $O(\sqrt{\log(T)})$ (in both the gap-dependent and independent bounds), to match the single-agent $\widetilde{O}(\sqrt{dMT})$ rate. Similarly, following the reasoning in [48], we can show that with $O(M^{1.5}d^3)$ communication (i.e., independent of $T$), we incur regret of $O(\sqrt{dMT}\log^2(MT))$.*

Theorem 1 demonstrates the relationship between communication complexity (i.e., number of synchronization rounds) and the regret bound for a fixed privacy budget, via the dependence on the bounds $\rho_{\min}, \rho_{\max}$ and $\kappa$. We now present similar results (for a fixed privacy budget) on group regret for the decentralized setting, which is more involved, as the delays and lack of a centralized controller make it difficult to control the heterogeneity of information between agents. Subsequently, in the next section, we will present results on the privacy guarantees of our algorithms.

## 4.2 Decentralized Peer-to-Peer Environment

In this environment, we assume that the collection of agents communicate by directly sending messages to each other. The communication network is denoted by an undirected (connected) graph $\mathcal{G} = (V, E)$, where, edge $e_{ij} \in E$ if agents $i$ and $j$ can communicate directly. The protocol operates as follows: every trial, each agent interacts with their respective bandit, and obtains a reward. At any trial $t$, after receiving the reward, each agent $v$ sends the message $\boldsymbol{m}_{v,t}$ to all its neighbors in $\mathcal{G}$. This message is forwarded from agent to agent $\gamma$ times (taking one trial of the bandit problem each between forwards), after which it is dropped. This communication protocol, based on the *time-to-live* (delay) parameter $\gamma \leq \text{diam}(\mathcal{G})$ is a common technique to control communication complexity, known as the LOCAL protocol [19, 34, 42]. Each agent $v \in V$ therefore also receives messages $\boldsymbol{m}_{v',t-d(v,v')}$ from all the agents $v'$ such that $d(v, v') \leq \gamma$, i.e., from all agents in $N_\gamma(v)$.

There are several differences in this setting compared to the previous one: first, since agents receive messages from different other agents based on their position in $\mathcal{G}$, they generally have heterogenous information throughout (no global sync). Second, information does not flow instantaneously through $\mathcal{G}$, and messages can take up to $\gamma$ rounds to be communicated (delay). This requires (mainly technical) changes to the centralized algorithm in order to control the regret. We present the pseudocode our algorithm DECENTRALIZED FEDUCB in the Appendix, but highlight the major differences from the centralized version as follows. The first key algorithmic change from the earlier variant is the idea of subsampling, inspired by the work of Weinberger and Ordentlich [50]. Each agent $i$ maintains $\gamma$ total estimators $\bar{\boldsymbol{\theta}}_{i,g}, g = 1, ..., \gamma$, and uses each estimator in a round-robin fashion, i.e., at $t = 1$ each agent uses $\bar{\boldsymbol{\theta}}_{i,1}$, and at $t = 2$, each agent uses $\bar{\boldsymbol{\theta}}_{i,2}$, and so on. These estimators (and their associated

---

**Algorithm 2** PRIVATIZER$(\varepsilon, \delta, M, T)$ for any agent $i$

---

1: **Initialization:**
2: If communication rounds $n$ are fixed *a priori*, set $m \leftarrow 1 + \lceil \log n \rceil$, else $m \leftarrow 1 + \lceil \log T \rceil$.
3: Create binary tree $\mathcal{T}$ of depth $m$.
4: **for** node $n$ in $\mathcal{T}$ **do**
5:      Sample noise $\widehat{\boldsymbol{N}} \in \mathbb{R}^{(d+1) \times (d+1)}$, where $\widehat{\boldsymbol{N}}_{ij} \sim \mathcal{N}\left(0, 16m(L^2+1)^2 \log(2/\delta)^2/\varepsilon^2\right)$.
6:      Store $\boldsymbol{N} = (\widehat{\boldsymbol{N}} + \widehat{\boldsymbol{N}}^\top)/\sqrt{2}$ at node $n$.
7: **end for**
8: **Runtime:**
9: **for** each communication round $t \le n$ **do**
10:      Receive $\widehat{\boldsymbol{Q}}_{i,t}$ from agent, and insert it into $\mathcal{T}$ (see [26], Alg. 5).
11:      Compute $\boldsymbol{M}_{i,t+1}$ using the least nodes of $\mathcal{T}$ (see [26], Alg. 5).
12:      Set $\widehat{\boldsymbol{U}}_{i,t+1} = \boldsymbol{U}_{i,t+1} + \boldsymbol{H}_{i,t}$ as top-left $d \times d$ submatrix of $\boldsymbol{M}_{i,t+1}$.
13:      Set $\widehat{\boldsymbol{u}}_{i,t+1} = \boldsymbol{u}_{i,t+1} + \boldsymbol{h}_{i,t}$ as first $d$ entries of last column of $\boldsymbol{M}_{i,t+1}$.
14:      Return $\widehat{\boldsymbol{U}}_{i,t+1}, \widehat{\boldsymbol{u}}_{i,t+1}$ to agent.
15: **end for**

---

$\boldsymbol{V}_{i,g}, \tilde{\boldsymbol{u}}_{i,g})$ are updated in a manner similar to Alg. 1: if the $\log \det$ of the $g^{th}$ Gram matrix exceeds a threshold $D/(\Delta_{i,g} + 1)$, then the agent broadcasts a request to synchronize. Each agent $j$ within the $\gamma$-clique of $i$ that receives this request broadcasts its own $\boldsymbol{V}_{j,g}, \tilde{\boldsymbol{u}}_{j,g}$. Therefore, each $\boldsymbol{V}_{i,g}, \tilde{\boldsymbol{u}}_{i,g}$ is updated with action/reward pairs from *only* the trials they were employed in (across all agents). This ensures that if a signal to synchronize the $g^{th}$ set of parameters has been broadcast by an agent $i$, all agents within the $\gamma$-clique of $i$ will have synchronized their $g^{th}$ parameters by the next 2 rounds they will be used again (i.e., $2\gamma$ trials later). We defer most of the intermediary results to the Appendix for brevity, and present the primary regret bound (in terms of privacy parameters $\rho_{\min}, \rho_{\max}$ and $\kappa$):

**Theorem 2** (Decentralized Private Group Regret). *Assuming Proposition 2 holds, and synchronization occurs in at least $n = \Omega\left(d(\bar{\chi}(\mathcal{G}_\gamma) \cdot \gamma)(1 + L^2)^{-1} \log\left(\rho_{\max}/\rho_{\min} + TL^2/d\rho_{\min}\right)\right)$ rounds, decentralized FEDUCB obtains the following group pseudoregret with probability at least $1 - \alpha$:*

$$\mathcal{R}_M(T) = O\left(\sigma\sqrt{M(\bar{\chi}(\mathcal{G}_\gamma) \cdot \gamma)Td}\left(\log\left(\frac{\rho_{\max}}{\rho_{\min}} + \frac{TL^2}{\gamma d\rho_{\min}}\right) + \sqrt{\log\frac{2}{\alpha}} + S\sqrt{M\rho_{\max}} + \kappa\sqrt{M}\right)\right).$$

**Remark 2** (Decentralized Group Regret). *Decentralized FEDUCB obtains an identical dependence on the privacy bounds $\rho_{\min}, \rho_{\max}$ and $\kappa$ and horizon $T$ as Algorithm 1, since the underlying bandit subroutines are identical for both. The key difference is in additional the leading factor of $\sqrt{\bar{\chi}(\mathcal{G}_\gamma) \cdot \gamma}$, which arises from the delayed spread of information: if $\mathcal{G}$ is dense, e.g., complete, then $\gamma = 1$ and $\bar{\chi}(\mathcal{G}_\gamma) = 1$, since there is only one clique of $\mathcal{G}$. In the worst case, if $\mathcal{G}$ is a line graph, then $\bar{\chi}(\mathcal{G}_\gamma) = M/\gamma$, giving an additional factor of $M$ (i.e., it is as good as each agent acting individually). In more practical scenarios, we expect $\mathcal{G}$ to be hierarchical, and expect a delay overhead of $o(1)$. Note that since each agent only communicates within its clique, the factor $\bar{\chi}(\mathcal{G}_\gamma)$ is optimal, see [12].*

### 4.3 Privacy Guarantees

We now discuss the privacy guarantees for both algorithms. Here we present results for the centralized algorithm, but our results hold (almost identically) for the decentralized case as well (see appendix). Note that each agent interacts with data from other agents only via the cumulative parameters $\boldsymbol{S}_t$ and $\boldsymbol{s}_t$. These, in turn, depend on $\boldsymbol{Z}_{i,t} = \boldsymbol{U}_{i,t} + \boldsymbol{H}_{i,t}$ and $\boldsymbol{z}_{i,t} = \bar{\boldsymbol{u}}_{i,t} + \boldsymbol{h}_{i,t}$ for each agent $i$, on the instances $t$ that synchronization occurs.

**Proposition 3** (see [16, 41]). *Consider $n \le T$ synchronization rounds occuring on trials $\bar{\boldsymbol{\sigma}} \subseteq [T]$. If the sequence $(\boldsymbol{Z}_{i,t}, \boldsymbol{z}_{i,t})_{t \in \bar{\boldsymbol{\sigma}}}$ is $(\varepsilon, \delta)$-differentially private with respect to $(\boldsymbol{x}_{i,t}, y_{i,t})_{t \in [T]}$, for each agent $i \in [M]$, then all agents are $(\varepsilon, \delta, M)$-federated differentially private.*

**Tree-Based Mechanism**. Let $x_1, x_2, ... x_T$ be a (matrix-valued) sequence of length $T$, and $s_i = \sum_{t=1}^{i} x_t$ be the incremental sum of the sequence that must be released privately. The tree-based mechanism [17] for differential privacy involves a trusted entity maintaining a binary tree $\mathcal{T}$ (of depth $m = 1 + \lceil \log_2 T \rceil$), where the leaf nodes contain the sequence items $x_i$, and each parent node maintains the (matrix) sum of its children. Let $n_i$ be the value stored at any node in the tree. The

mechanism achieves privacy by adding a noise $h_i$ to each node, and releasing $n_i + h_i$ whenever a node is queried. Now, to calculate $s_t$ for some $t \in [T]$, the procedure is to traverse the tree $\mathcal{T}$ up to the leaf node corresponding to $x_t$, and summing up the values at each node on the traversal path. The advantage is that we only access at most $m$ nodes, and add $m = O(\log T)$ noise (instead of $O(T)$).

The implementation of the private release of $(\boldsymbol{U}_{i,t}, \bar{\boldsymbol{u}}_{i,t})$ is done by the ubiquitous tree-based mechanism for partial sums. Following [41], we also aggregate both into a single matrix $\boldsymbol{M}_{i,t} \in \mathbb{R}^{(d+1)\times(d+1)}$, by first concatenating: $\boldsymbol{A}_{i,t} := [\boldsymbol{X}_{i,1:t}, \boldsymbol{y}_{i,1:t}] \in \mathbb{R}^{t\times(d+1)}$, and then computing $\boldsymbol{M}_{i,t} = \boldsymbol{A}_{i,t}^\top \boldsymbol{A}_{i,t}$. Furthermore, the update is straightforward: $\boldsymbol{M}_{i,t+1} = \boldsymbol{M}_{i,t} + [\boldsymbol{x}_{i,t}^\top \ y_{i,t}]^\top [\boldsymbol{x}_{i,t}^\top \ y_{i,t}]$. Recall that in our implementation, we only communicate in synchronization rounds (and not every round). Assume that two successive rounds of synchronization occur at time $t'$ and $t$. Then, at instant $t$, each agent $i$ inserts $\sum_{\tau=t'}^{t} [\boldsymbol{x}_{i,\tau}^\top \ y_{i,\tau}]^\top [\boldsymbol{x}_{i,\tau}^\top \ y_{i,\tau}]$ into $\mathcal{T}$, and computes $(\boldsymbol{U}_{i,t}, \bar{\boldsymbol{u}}_{i,t})$ by summing up the entire path up to instant $t$ via the tree mechanism. Therefore, the tree mechanism accesses at most $m = 1 + \lceil \log_2 n \rceil$ nodes (where $n$ total rounds of communication occur until instant $T$), and hence noise that ensures each node guarantees $(\varepsilon/\sqrt{8m\ln(2/\delta)}, \delta/2m)$-privacy is sufficient to make the outgoing sequence $(\varepsilon, \delta)$-private. This is different from the setting in the joint DP single-agent bandit [41], where observations are inserted *every* round, and not only for synchronization rounds.

To make the partial sums private, a noise matrix is also added to each node in $\mathcal{T}$. We utilize additive Gaussian noise: at each node, we sample $\widehat{\boldsymbol{N}} \in \mathbb{R}^{(d+1)\times(d+1)}$, where each $\widehat{\boldsymbol{N}}_{i,j} \sim \mathcal{N}(0, \sigma_N^2)$, and $\sigma_N^2 = 16m(L^2+1)^2 \log(2/\delta)^2/\varepsilon^2$, and symmetrize it (see step 6 of Alg. 2). The total noise $\boldsymbol{H}_{i,t}$ is the sum of at most $m$ such terms, hence the variance of each element in $\boldsymbol{H}_{i,t}$ is $\leq m\sigma_N^2$. We can bound the operator norm of the top-left $(d \times d)$-submatrix of each noise term. Therefore, to guarantee $(\varepsilon, \delta, M)$-federated DP, we require that with probability at least $1 - \alpha/nM$:

$$\|\boldsymbol{H}_{i,t}\|_{\mathrm{op}} \leq \Lambda = \sqrt{32}m(L^2+1)\log(4/\delta)(4\sqrt{d} + 2\ln(2nM/\alpha))/\varepsilon.$$

**Remark 3** (Privacy Guarantee). *The procedure outlined above guarantees that each of the $n$ outgoing messages $(\boldsymbol{U}_{i,t}, \bar{\boldsymbol{u}}_{i,t})$ (where $t$ is a synchronization round) for any agent $i$ is $(\varepsilon, \delta)$-differentially private. This analysis considers the $L_2$-sensitivity with respect to a single differing observation, i.e., $(\boldsymbol{x}, y)$ and not the entire message itself, i.e., the complete sequence $(\boldsymbol{x}_{i,\tau}, y_{i,\tau})_{\tau=t'}^{t}$ (where $t'$ and $t$ are successive synchronization rounds), which may potentially have $O(t)$ sensitivity and warrants a detailed analysis. While our analysis is sufficient for the user-level adversary model, there may be settings where privacy is required at the message-level as well, which we leave as future work.*

However, as noted by [41], this $\boldsymbol{H}_{i,t}$ would not always be PSD. To ensure that it is always PSD, we can shift each $\boldsymbol{H}_{i,t}$ by $2\Lambda\boldsymbol{I}$, giving a bound on $\rho_{\max}$. Similarly, we can obtain bounds on $\rho_{\min}$ and $\kappa$:

**Proposition 4.** *Fix $\alpha > 0$. If each agent $i$ samples noise parameters $\boldsymbol{H}_{i,t}$ and $\boldsymbol{h}_{i,t}$ using the tree-based Gaussian mechanism mentioned above for all $n$ trials of $\bar{\boldsymbol{\sigma}}$ in which communication occurs, then the following $\rho_{\min}, \rho_{\max}$ and $\kappa$ are $(\alpha/2nM, \bar{\boldsymbol{\sigma}})$-accurate bounds:*

$$\rho_{\min} = \Lambda, \ \rho_{\max} = 3\Lambda, \ \kappa \leq \sqrt{m(L^2+1)\left(\sqrt{d} + 2\log(2nM/\alpha)\right)}/(\sqrt{2}\varepsilon).$$

**Remark 4** (Strategyproof Analysis). *The mechanism presented above assumes the worst-case communication, i.e., synchronization occurs every round, therefore at most $T$ partial sums will be released, and $m = 1 + \lceil \log T \rceil$. This is not true for general settings, where infrequent communication would typically require $m = O(\log \log T)$. However, if any agent is byzantine and requests a synchronization every trial, $m$ must be $1 + \lceil \log T \rceil$ to ensure privacy. In case the protocol is fixed in advance (i.e., synchronization occurs on a pre-determined set $\bar{\boldsymbol{\sigma}}$ of $n$ rounds), then we can set $m = 1 + \lceil \log n \rceil$ to achieve the maximum utility at the desired privacy budget.*

**Remark 5** (Decentralized Protocol). *Decentralized FEDUCB obtains similar bounds for $\rho_{\min}, \rho_{\max}$ and $\kappa$, with $m = 1 + \lceil \log(T/\gamma) \rceil$. An additional term of $\log(\gamma)$ appears in $\Lambda$ and $\kappa$, since we need to now maintain $\gamma$ partial sums with at most $T/\gamma$ elements in the worst-case (see Appendix). Unsurprisingly, there is no dependence on the network $\mathcal{G}$, as privatization is done at the source itself.*

**Corollary 1** (($\varepsilon, \delta$)-dependent Regret). *FEDUCB with the PRIVATIZER subroutine in Alg. 2, obtains $\widetilde{O}(d^{3/4}\sqrt{MT/\varepsilon})$ centralized regret and $\widetilde{O}(d^{3/4}\sqrt{(\bar{\chi}(\mathcal{G}_\gamma) \cdot \gamma)MT/\varepsilon})$ decentralized regret.*

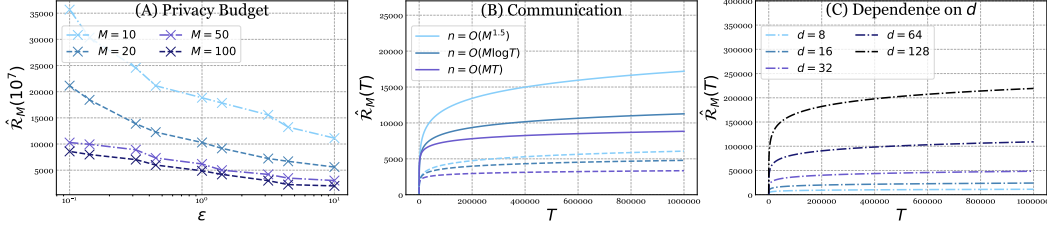

Figure 1: A comparison of centralized FEDUCB on 3 different axes. Fig. (A) describes the variation in asymptotic per-agent regret for varying privacy budget $\varepsilon$ (where $\delta = 0.1$); (B) describes the effect of $n$ in private (solid) vs. non-private (dashed) settings; (C) describes the effect of $d$ in per-agent regret in the private setting ($n = O(M \log T), \varepsilon = 1, \delta = 0.1$). Experiments averaged over 100 runs.

## 4.4 Experiments

We provide an abridged summary of our experiments (ref. Appendix for all experiments). Here, we focus on the centralized environment, and on the variation of the regret with communication complexity and privacy budget. For all experiments, we assume $L = S = 1$. For any $d$, we randomly fix $\boldsymbol{\theta}^* \in \mathcal{B}_d(1)$. Each $\mathcal{D}_{i,t}$ is generated as follows: we randomly sample $K \leq d^2$ actions $\boldsymbol{x}$, such that for $K - 1$ actions $0.5 \leq \langle \boldsymbol{x}, \boldsymbol{\theta}^* \rangle \leq 0.6$ and for the optimal $\boldsymbol{x}^*, 0.7 \leq \langle \boldsymbol{x}^*, \boldsymbol{\theta}^* \rangle \leq 0.8$ such that $\Delta \geq 0.1$ always. $y_{i,t}$ is sampled from $\mathrm{Ber}(\langle \boldsymbol{x}_{i,t}, \boldsymbol{\theta}^* \rangle)$ such that $\mathbb{E}[y_{i,t}] = \langle \boldsymbol{x}_{i,t}, \boldsymbol{\theta}^* \rangle$ and $|y_{i,t}| \leq 1$. Results are in Fig. 1, and experiments are averaged on 100 trials.

**Experiment 1: Privacy Budget**. In this setting, we set $n = O(M \log T), d = 10$ (to balance communication and performance), and plot the average per-agent regret after $T = 10^7$ trials for varying $M$ and $\varepsilon$, while keeping $\delta = 0.1$. Figure 1A describes the results, competitive even at large privacy budget.

**Experiment 2: Communication**. To test the communication-privacy trade-off, we plot the regret curves for $M = 100$ agents on $n = O(M^{1.5}), O(M \log T), O(MT)$ communication for both private ($\varepsilon = 1$) and non-private settings. We observe a tradeoff as highlighted in Remark 1.

**Experiment 3: Dependence on** $d$. Finally, as an ablation, we provide the average per-agent regret curves for $M = 100$ by varying the dimensionality $d$. We observe essentially a quadratic dependence.

## 5 Discussion and Conclusion

The relentless permeance of intelligent decision-making on sensitive user data necessitates the development of technically sound and robust machine learning systems; it is difficult to stress enough the importance of protecting user data in modern times. There is a significant effort [7, 22] in creating decentralized decision-making systems that benefit from pooling data from multiple sources while maintaining user-level privacy. This research provides an instance of such a system, along with guarantees on both its utility and privacy. From a technical perspective, we make improvements along several fronts – while there has been prior work on multi-agent private linear bandits [21, 38] our work is the first to provide rigorous guarantees on private linear bandits in the multi-agent setting. Our work additionally provides the first algorithm with regret guarantees for contextual bandits in decentralized networks, extending the work of many on multi-armed bandits [39, 31, 30, 14, 12].

There are several unresolved questions in this line of work, as highlighted by our algorithm itself. In the decentralized case, our algorithm obtains a communication overhead of $O(\sqrt{\bar{\chi}(\mathcal{G}_\gamma) \cdot \gamma})$, which we comment is an artefact of our proof technique. We conjecture that the true dependence scales as $O(\sqrt{\alpha(\mathcal{G}_\gamma) + \gamma})$ (where $\alpha(\cdot)$ is the independence number of the graph), that would be obtained by an algorithm that bypasses subsampling, with a more sophisticated synchronization technique. Additionally, the lower bound obtained for private contextual bandit estimation [41] has an $O(\varepsilon^{-1})$ dependence on the privacy budget. In the federated setting, given the intricate relationship between communication and privacy, an interesting line of inquiry would be to understand the communication-privacy tradeoff in more detail, with appropriate lower bounds for fixed communication budgets.

## Broader Impact

We envision a world that runs on data-dependent inference. Given the ease of availability of sensitive data, large-scale centralized data sources are not feasible to uphold the best interests and safety of the average user, and decentralized inference mechanisms are a feasible alternative balancing utility with important normative values as privacy, transparency and accountability. This research is a part of such a broader research goal, to create decentralized systems that maintain the security of the end-user in mind. We believe that methods like ours are a starting point for better private algorithms that can be readily deployed in industry, with appropriate guarantees as well. While there are several subsequent steps that can be improved both theoretically and practically, the design for this research from the ground-up has been to ensure robustness and security (sometimes) in lieu of efficiency.

## Funding Disclosures

This project was funded by the MIT Trust:Data Consortium.

## Footnotes

[1]Originally, federated learning referred to the algorithm proposed in [28] for supervised learning, however, now the term broadly refers to the distributed cooperative learning setting [27].

[2]The *pseudoregret* is an expectation (over the randomness of $\eta_{i,t}$) of the stochastic quantity *regret*, and is more amenable to high-probability bounds. However, a bound over the pseudoregret can also bound the regret with high probability, e.g., by a Hoeffding concentration (see, e.g., [47]).

[3]Assuming bounded rewards is required for the privacy mechanism, and is standard in the literature [3, 2].

[4]Under the stronger assumption that each agent interacts with a completely different set of individuals, we do not need to invoke the composition theorem (as $\boldsymbol{x}_{1,t}, \boldsymbol{x}_{2,t}, ..., \boldsymbol{x}_{M,t}$ are independent for each $t$). However, in the case that one individual could potentially interact simultaneously with all agents, this is not true (e.g., when for some $t$, $\mathcal{D}_{i,t} = \mathcal{D}_{j,t} \; \forall i, j$) and we must invoke the $k$-fold composition Theorem[17] to ensure privacy.

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
