[Supplementary Material]

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

*Proof.* We denote the number of (common) bandit trials between two rounds of communication as an epoch. Let $n' = \sqrt{\frac{DT}{d\log(\rho_{\max}/\rho_{\min} + TL^2/(d\rho_{\min}))}} + 1$. There can be at most $\lceil T/n' \rceil$ rounds of communication such that they occur after an epoch of length $n'$. On the other hand, if there is any round of communication succeeding an epoch (that begins, say at time $t$) of length $< n'$, then for that epoch, $\log \frac{\det(\boldsymbol{S}_{i,t+n'})}{\det(\boldsymbol{S}_{i,t})} > D/n'$. Let the communication occur at a set of rounds $t'_1, ..., t'_n$. Now, since:

$$\sum_{i=1}^{n-1} \log \frac{\det\left(\boldsymbol{S}_{i,t'_{i+1}}\right)}{\det\left(\boldsymbol{S}_{i,t_i}\right)} = \log \frac{\det\left(\boldsymbol{S}_{i,T}\right)}{\det\left(\boldsymbol{S}_{i,0}\right)} \leq d\log(\rho_{\max}/\rho_{\min} + TL^2/(d\rho_{\min})), \qquad (4)$$

We have that the total number of communication rounds succeeding epochs of length less than $n'$ is upper bounded by $\log \frac{\det(\boldsymbol{S}_{i,T})}{\det(\boldsymbol{S}_{i,0})} \leq d\log(\rho_{\max}/\rho_{\min} + TL^2/(d\rho_{\min})) \cdot (n'/D)$. Combining both the results together, we have the total rounds of communication as:

$$n \leq \lceil T/n' \rceil + \lceil d\log(\rho_{\max}/\rho_{\min} + TL^2/(d\rho_{\min})) \cdot (n'/D) \rceil \qquad (5)$$

$$\leq T/n' + d\log(\rho_{\max}/\rho_{\min} + TL^2/(d\rho_{\min})) \cdot (n'/D) + 2 \qquad (6)$$

Replacing $n'$ from earlier gives us the final result. $\qquad \square$

**Proposition 6** ($\beta_{i,t}$, Proposition 2 of main paper)**.** *Consider an instance of the problem where synchronization occurs exactly $n$ times on instances $\bar{\boldsymbol{\sigma}}$, up to and including $T$ trials, and $\rho_{\min}, \rho_{\max}$ and $\kappa$ are $(\alpha/2nM)$-accurate. Then, for Algorithm 1, the sequence $(\beta_{i,t})_{i\in[M],t\in[T]}$ is $(\alpha, M, T)$-accurate where:*

$$\beta_{i,t} := \sigma\sqrt{2\log(^2/\alpha) + d\log\left(\det(\boldsymbol{V}_{i,t})\right) - d\log(M\rho_{\min})} + S\sqrt{M\rho_{\max}} + \kappa\sqrt{M}.$$

*Proof.* Let $\boldsymbol{X}_{i,<t}$ denote the set of all observations available to the agent (including private communication from other agents). Furthermore, let the noise-free Gram matrix of all observations as $\boldsymbol{G}_{i,t} = \sum_{\tau=1}^{t} \boldsymbol{x}_{i,\tau}\boldsymbol{x}_{i,\tau}^{\top} + \sum_{\tau=1}^{t_s} \sum_{j=1,j\neq i}^{M} \boldsymbol{x}_{j,\tau}\boldsymbol{x}_{j,\tau}^{\top}$ (where $t_s$ is the last synchronization iteration). We also have that $\boldsymbol{V}_{i,t} = \boldsymbol{G}_{i,t} + \sum_{j=1}^{M} \boldsymbol{H}_{j,t}$, and for any $t$ between synchronization rounds $t_s$ and $t_{s+1}$, $\boldsymbol{H}_{j,t} = \boldsymbol{H}_{j,t_s}$ for all $j$. By definition, $\tilde{\boldsymbol{\theta}}_{i,t} = \boldsymbol{V}_{i,t}^{-1}\tilde{\boldsymbol{u}}_{i,t}$, $\tilde{\boldsymbol{u}}_{t,i} = \boldsymbol{u}_{i,t} + \sum_{j\in[M]} \boldsymbol{h}_{j,t}$ and $\boldsymbol{u}_{i,t} = \boldsymbol{X}_{i,<t}^{\top}\boldsymbol{y}_{<t}$. Therefore, we have that,

$$\boldsymbol{\theta}^* - \tilde{\boldsymbol{\theta}}_{i,t} = \boldsymbol{\theta}^* - \boldsymbol{V}_{i,t}^{-1}\left(\boldsymbol{X}_{i,<t}^{\top}\boldsymbol{y}_{<t} + \sum_{j\in[M]} \boldsymbol{h}_{j,t}\right)$$

$$= \boldsymbol{\theta}^* - \boldsymbol{V}_{i,t}^{-1}\left(\boldsymbol{X}_{i,<t}^{\top}\boldsymbol{X}_{i,<t}\boldsymbol{\theta}^* + \boldsymbol{X}_{i,<t}^{\top}\boldsymbol{\eta}_{<t} + \sum_{j\in[M]} \boldsymbol{h}_{j,t}\right)$$

$$= \boldsymbol{\theta}^* - \boldsymbol{V}_{i,t}^{-1}\left(\boldsymbol{V}_{i,t}\boldsymbol{\theta}^* - \sum_{j\in[M]} \boldsymbol{H}_{j,t}\boldsymbol{\theta}^* + \boldsymbol{X}_{i,<t}^{\top}\boldsymbol{\eta}_{<t} + \sum_{j\in[M]} \boldsymbol{h}_{j,t}\right)$$

$$= \boldsymbol{V}_{i,t}^{-1}\left(\sum_{j\in[M]} \boldsymbol{H}_{j,t}\boldsymbol{\theta}^* - \boldsymbol{X}_{i,<t}^{\top}\boldsymbol{\eta}_{<t} - \sum_{j\in[M]} \boldsymbol{h}_{j,t}\right).$$

Multiplying both sides by $V_{i,t}^{1/2}$ gives

$$V_{i,t}^{1/2}\left(\boldsymbol{\theta}^* - \tilde{\boldsymbol{\theta}}_{i,t}\right) = V_{i,t}^{-1/2}\left(\sum_{j\in[M]} \boldsymbol{H}_{j,t}\boldsymbol{\theta}^* - \boldsymbol{X}_{i,<t}^\top\boldsymbol{\eta}_{<t} - \sum_{j\in[M]} \boldsymbol{h}_{j,t}\right)$$

$$\implies \left\|\boldsymbol{\theta}^* - \tilde{\boldsymbol{\theta}}_{i,t}\right\|_{V_{i,t}} = \left\|\sum_{j\in[M]} \boldsymbol{H}_{j,t}\boldsymbol{\theta}^* - \boldsymbol{X}_{i,<t}^\top\boldsymbol{\eta}_{<t} - \sum_{j\in[M]} \boldsymbol{h}_{j,t}\right\|_{V_{i,t}^{-1}} \qquad \text{(Applying } \|\cdot\| \text{)}$$

$$\leq \left\|\sum_{j\in[M]} \boldsymbol{H}_{j,t}\boldsymbol{\theta}^*\right\|_{V_{i,t}^{-1}} + \left\|\boldsymbol{X}_{i,<t}^\top\boldsymbol{\eta}_{<t}\right\|_{V_{i,t}^{-1}} + \left\|\sum_{j\in[M]} \boldsymbol{h}_{j,t}\right\|_{V_{i,t}^{-1}}$$

$$\text{(Triangle inequality)}$$

Making the substitution $\boldsymbol{H}_t = \sum_{j\in[M]} \boldsymbol{H}_{j,t}$,

$$\leq \|\boldsymbol{H}_t\boldsymbol{\theta}^*\|_{\boldsymbol{H}_t^{-1}} + \left\|\boldsymbol{X}_{i,<t}^\top\boldsymbol{\eta}_{<t}\right\|_{V_{i,t}^{-1}} + \|\boldsymbol{h}_{j,t}\|_{\boldsymbol{H}_t^{-1}}$$

$$\text{(Since } V_{i,t} \succcurlyeq \sum_{j\in[M]} \boldsymbol{H}_{j,t}\text{)}$$

$$\leq \|\boldsymbol{\theta}^*\|_{\boldsymbol{H}_t} + \left\|\boldsymbol{X}_{i,<t}^\top\boldsymbol{\eta}_{<t}\right\|_{(\boldsymbol{G}_{i,t}+M\rho_{\min}\boldsymbol{I})^{-1}} + \sum_{j\in[M]} \|\boldsymbol{h}_{j,t}\|_{\boldsymbol{H}_t^{-1}}$$

$$\text{(Since } \forall i \in [M], V_{i,t} \succcurlyeq \boldsymbol{G}_{i,t} + M\rho_{\min}\boldsymbol{I}\text{)}$$

Now, note that since we only require at most $Mn$ different noise matrices, we only need the noise sequences $\boldsymbol{H}$ by a union bound over all $TM$ rounds ($T$ rounds per agent), we can say that simultaneously for all $i \in [M], t \in [T]$, with probability at least $1 - \alpha/2$, $\|\boldsymbol{\theta}^*\|_{\boldsymbol{H}_t} \leq \sqrt{\|\boldsymbol{H}_t\|}\|\boldsymbol{\theta}^*\| \leq S\sqrt{M\rho_{\max}}$ and $\|\sum_{j\in[M]} \boldsymbol{h}_{i,t}\|_{\boldsymbol{H}_{i,t}^{-1}} \leq \kappa\sqrt{M}$. Next, by Theorem 1 of Abbasi-Yadkori *et al.* [1], we have that, with probability at least $1 - \alpha/2$ simultaneously,

$$\left\|\boldsymbol{X}_{i,<t}^\top\boldsymbol{\eta}_{<t}\right\|_{(\boldsymbol{G}_{i,t}+M\rho_{\min}\boldsymbol{I})^{-1}} \leq \sigma\sqrt{2\log\frac{2}{\alpha} + \log\frac{\det(\boldsymbol{G}_{i,t} + M\rho_{\min}\boldsymbol{I})}{\det(M\rho_{\min}\boldsymbol{I})}}$$

$$\leq \sigma\sqrt{2\log\frac{2}{\alpha} + d\log\left(\frac{\rho_{\max}}{\rho_{\min}} + \frac{tL^2}{d\rho_{\min}}\right)}.$$

The last step follows from (a) noting that $\forall i \in [M], \boldsymbol{G}_{i,t} + M\rho_{\max}\boldsymbol{I} \succcurlyeq V_{i,t} \succcurlyeq \boldsymbol{G}_{i,t} + M\rho_{\min}\boldsymbol{I} \implies \det(\boldsymbol{G}_{i,t} + M\rho_{\max}\boldsymbol{I}) \geq \det(V_{i,t}) \geq \det(\boldsymbol{G}_{i,t} + M\rho_{\min}\boldsymbol{I})$, and (b) the trace-determinant inequality. Putting it all together, we have that all $\beta_{i,t}$ are bounded by $\bar{\beta}_t$, given by,

$$\bar{\beta}_t := \sigma\sqrt{2\log\frac{2}{\alpha} + d\log\left(\frac{\rho_{\max}}{\rho_{\min}} + \frac{tL^2}{d\rho_{\min}}\right)} + S\sqrt{M\rho_{\max}} + \kappa\sqrt{M}.$$

$\square$

**Lemma 1.** *The instantaneous pseudoregret $r_{i,t}$ obtained by any agent $i$ at any instant $t$ obeys the following:*

$$r_{i,t} \leq 2\bar{\beta}_T \|\boldsymbol{x}_{i,t}\|_{V_{i,t}^{-1}}$$

*Proof.* At every round, each agent $i$ selects an "optimistic" action $\boldsymbol{x}_{i,t}$ such that,

$$\left(\boldsymbol{x}_{i,t}, \bar{\boldsymbol{\theta}}_{i,t}\right) = \underset{(\boldsymbol{x},\boldsymbol{\theta})\in\mathcal{D}_{i,t}\times\mathcal{E}_{i,t}}{\arg\max} \langle\boldsymbol{x}, \boldsymbol{\theta}\rangle. \tag{7}$$

Let $\boldsymbol{x}_{i,t}^*$ be the optimal action at time $t$ for agent $i$, i.e., $\boldsymbol{x}_{i,t}^* = \arg\max_{\boldsymbol{x}\in\mathcal{D}_{i,t}} \langle \boldsymbol{x}, \boldsymbol{\theta}^*\rangle$. We can then decompose the immediate pseudoregret $r_{i,t}$ for agent $i$ as the following.

$$
\begin{aligned}
r_{i,t} &= \langle \boldsymbol{x}_{i,t}^*, \boldsymbol{\theta}^*\rangle - \langle \boldsymbol{x}_{i,t}, \boldsymbol{\theta}^*\rangle \\
&\leq \langle \boldsymbol{x}_{i,t}, \bar{\boldsymbol{\theta}}_{i,t}\rangle - \langle \boldsymbol{x}_{i,t}, \boldsymbol{\theta}^*\rangle && \text{(Since } (\boldsymbol{x}_{i,t}, \bar{\boldsymbol{\theta}}_{i,t}) \text{ is optimistic)} \\
&= \langle \boldsymbol{x}_{i,t}, \bar{\boldsymbol{\theta}}_{i,t} - \boldsymbol{\theta}^*\rangle \\
&= \left\langle \boldsymbol{V}_{i,t}^{-1/2}\boldsymbol{x}_{i,t}, \boldsymbol{V}_{i,t}^{1/2}\left(\bar{\boldsymbol{\theta}}_{i,t} - \boldsymbol{\theta}^*\right)\right\rangle && (\boldsymbol{V}_{i,t} \succcurlyeq 0) \\
&\leq \|\boldsymbol{x}_{i,t}\|_{\boldsymbol{V}_{i,t}^{-1}} \left\|\bar{\boldsymbol{\theta}}_{i,t} - \boldsymbol{\theta}^*\right\|_{\boldsymbol{V}_{i,t}} && \text{(Cauchy-Schwarz)} \\
&\leq \|\boldsymbol{x}_{i,t}\|_{\boldsymbol{V}_{i,t}^{-1}} \left(\left\|\bar{\boldsymbol{\theta}}_{i,t} - \tilde{\boldsymbol{\theta}}_{i,t}\right\|_{\boldsymbol{V}_{i,t}} + \left\|\tilde{\boldsymbol{\theta}}_{i,t} - \boldsymbol{\theta}^*\right\|_{\boldsymbol{V}_{i,t}}\right) && \text{(Triangle inequality)} \\
&\leq 2\beta_{t,i}\|\boldsymbol{x}_{i,t}\|_{\boldsymbol{V}_{i,t}^{-1}} && \text{(Since } \bar{\boldsymbol{\theta}}_{i,t}, \boldsymbol{\theta}^* \in \mathcal{E}_{i,t}) \\
&\leq 2\bar{\beta}_T\|\boldsymbol{x}_{i,t}\|_{\boldsymbol{V}_{i,t}^{-1}}. && \text{(By Proposition 2)}
\end{aligned}
$$

$\square$

**Lemma 2** (Elliptical Potential, Lemma 3 of [1], Lemma 22 of [41]). *Let $\boldsymbol{x}_1, \boldsymbol{x}_2, ..., \boldsymbol{x}_n \in \mathbb{R}^d$ be vectors such that $\|\boldsymbol{x}\|_2 \leq L$. Then, for any positive definite matrix $\boldsymbol{U}_0 \in \mathbb{R}^{d\times d}$, define $\boldsymbol{U}_t := \boldsymbol{U}_{t-1} + \boldsymbol{x}_t\boldsymbol{x}_t^\top$ for all $t$. Then, for any $\nu > 1$,*

$$
\sum_{t=1}^n \|\boldsymbol{x}_t\|_{\boldsymbol{U}_{t-1}^{-1}}^2 \leq 2d\log_\nu\left(\frac{tr(\boldsymbol{U}_0) + nL^2}{d\det^{1/d}(\boldsymbol{U}_0)}\right).
$$

*Proof.* We urge the readers to refer to [41] for a proof of this statement. $\square$

**Theorem 3** (Private Group Regret with Centralized Controller, Theorem 1 of the main paper). *Assuming Proposition 2 holds, and synchronization occurs in at least $n = \Omega\left(d\log\left(\rho_{\max}/\rho_{\min} + TL^2/d\rho_{\min}\right)\right)$ rounds, Algorithm 1 obtains the following group pseudoregret with probability at least $1 - \alpha$:*

$$
\mathcal{R}_M(T) = O\left(\sigma\sqrt{MTd}\left(\log\left(\rho_{\max}/\rho_{\min} + TL^2/d\rho_{\min}\right) + \sqrt{\log 2/\alpha} + S\sqrt{M\rho_{\max}} + \kappa\sqrt{M}\right)\right).
$$

*Proof.* Consider a hypothetical agent that takes the following $MT$ actions $\boldsymbol{x}_{1,1}, \boldsymbol{x}_{2,1}, ..., \boldsymbol{x}_{1,2}, \boldsymbol{x}_{2,2}, ..., \boldsymbol{x}_{M-1,T}, \boldsymbol{x}_{M,T}$ sequentially. Let $\boldsymbol{W}_{i,t} = M\rho_{\min}\boldsymbol{I} + \sum_{j=1}^M \sum_{u=1}^{t-1} \boldsymbol{x}_{j,u}\boldsymbol{x}_{j,u}^\top + \sum_{j=1}^{i-1} \boldsymbol{x}_{j,t}\boldsymbol{x}_{j,t}^\top$ be the Gram matrix formed until the hypothetical agent reaches $\boldsymbol{x}_{i,t}$. By Lemma 2, we have that

$$
\sum_{t=1}^T \sum_{i=1}^M \|\boldsymbol{x}_{i,t}\|_{\boldsymbol{W}_{i,t}^{-1}}^2 \leq 2d\log\left(1 + \frac{TL^2}{d\rho_{\min}}\right). \tag{8}
$$

Now, in the original setting, let $T_1, T_2, ..., T_{p-1}$ be the trials at which synchronization occurs. After any round $T_k$ of synchronization, consider the cumulative Gram matrices of all observations obtained until that round as $\boldsymbol{V}_k, k = 1, ..., p-1$, regularized by $M\rho_{\min}\boldsymbol{I}$, i.e., $\boldsymbol{V}_k = \sum_{i\in[M]} \sum_{t=1}^{T_k} \boldsymbol{x}_{i,t}\boldsymbol{x}_{i,t}^\top + M\rho_{\min}\boldsymbol{I}$. Finally, let $\boldsymbol{V}_p$ denote the (regularized) Gram matrix with all trials at time $T$, and $\boldsymbol{V}_0 = M\rho_{\min}\boldsymbol{I}$. Therefore, we have that $\det(\boldsymbol{V}_0) = (M\rho_{\min})^d$, and that $\det(\boldsymbol{V}_p) \leq \left(\frac{\text{tr}(\boldsymbol{V}_p)}{d}\right)^d \leq (M\rho_{\max} + MTL^2/d)^d$. Therefore, for any $\nu > 1$,

$$
\log_\nu\left(\frac{\det(\boldsymbol{V}_p)}{\det(\boldsymbol{V}_0)}\right) \leq d\log_\nu\left(\frac{\rho_{\max}}{\rho_{\min}} + \frac{TL^2}{d\rho_{\min}}\right).
$$

Let $R = \left\lceil d\log_\nu\left(\frac{\rho_{\max}}{\rho_{\min}} + \frac{TL^2}{d\rho_{\min}}\right)\right\rceil$. It follows that in all but $R$ periods between synchronization,

$$
1 \leq \frac{\det(\boldsymbol{V}_k)}{\det(\boldsymbol{V}_{k-1})} \leq \nu. \tag{9}
$$

We consider the event $E$ to be the period $k$ when Equation 9 holds. Now, for any $T_{k-1} \le t \le T_k$, consider the immediate pseudoregret for any agent $i$. By Lemma 1, we have

$$
\begin{aligned}
r_{i,t} &\le 2\bar{\beta}_T \|x_{i,t}\|_{V_{i,t}^{-1}} \\
&\le 2\bar{\beta}_T \|x_{i,t}\|_{(G_{i,t}+M\rho_{\min}I)^{-1}} && (V_{i,t} \succcurlyeq G_{i,t}+M\rho_{\min}I) \\
&\le 2\bar{\beta}_T \|x_{i,t}\|_{W_{i,t}^{-1}} \cdot \sqrt{\frac{\det(W_{i,t})}{\det(G_{i,t}+M\rho_{\min}I)}} \\
&\le 2\bar{\beta}_T \|x_{i,t}\|_{W_{i,t}^{-1}} \cdot \sqrt{\frac{\det(V_k)}{\det(G_{i,t}+M\rho_{\min}I)}} && (V_k \succcurlyeq W_{i,t}) \\
&\le 2\bar{\beta}_T \|x_{i,t}\|_{W_{i,t}^{-1}} \cdot \sqrt{\frac{\det(V_k)}{\det(V_{k-1})}} && (G_{i,t}+M\rho_{\min}I \succcurlyeq V_{k-1}) \\
&\le 2\nu\bar{\beta}_T \|x_{i,t}\|_{W_{i,t}^{-1}}. && (\text{Event } E \text{ holds})
\end{aligned}
$$

Now, we can sum up the immediate pseudoregret over all such periods where $E$ holds to obtain the total regret for these periods. With probability at least $1-\alpha$,

$$
\begin{aligned}
\text{Regret}(T, E) &= \sum_{i=1}^{M} \sum_{t\in[T]: \text{E is true}} r_{i,t} \\
&\le \sqrt{MT \left( \sum_{i=1}^{M} \sum_{t\in[T]: \text{E is true}} r_{i,t}^2 \right)} \\
&\le 2\nu\bar{\beta}_T \sqrt{MT \left( \sum_{i=1}^{M} \sum_{t\in[T]: \text{E is true}} \|x_{i,t}\|_{W_{i,t}^{-1}} \right)} \\
&\le 2\nu\bar{\beta}_T \sqrt{MT \left( \sum_{i=1}^{M} \sum_{t\in[T]} \|x_{i,t}\|_{W_{i,t}^{-1}} \right)} \\
&\le 2\nu\bar{\beta}_T \sqrt{2MTd\log_\nu \left(1 + \frac{TL^2}{d\rho_{\min}}\right)}.
\end{aligned}
$$

Now let us consider the periods in which $E$ does not hold. In any such period between synchronization of length $t_k = T_k - T_{k-1}$, we have, for any agent $i$, the regret accumulated given by:

$$
\begin{aligned}
\text{Regret}([T_{k-1}, T_k]) &= \sum_{t=T_{k-1}}^{T_k} \sum_{i=1}^{M} r_{i,t} \\
&\le 2\nu\bar{\beta}_T \left( \sum_{i=1}^{M} \sqrt{t_k \sum_{t=T_{k-1}}^{T_k} \|x_{i,t}\|_{V_{i,t}^{-1}}^2} \right) \\
&\le 2\nu\bar{\beta}_T \left( \sum_{i=1}^{M} \sqrt{t_k \log_\nu \left( \frac{\det(V_{i,t+t_k})}{\det(V_{i,t})} \right)} \right) \\
&\le 2\nu\bar{\beta}_T \left( \sum_{i=1}^{M} \sqrt{t_k \log_\nu \left( \frac{\det(G_{i,t+t_k}+M\rho_{\max}I)}{\det(G_{i,t}+M\rho_{\min}I)} \right)} \right)
\end{aligned}
$$

By Algorithm 1, we know that for all agents, $t_k \log_\nu \left( \frac{\det(\boldsymbol{G}_{i,t+t_k} + M\rho_{\max}\boldsymbol{I})}{\det(\boldsymbol{G}_{i,t} + M\rho_{\min}\boldsymbol{I})} \right) \leq D$ (since there would be a synchronization round otherwise), therefore

$$\leq 2\nu\bar{\beta}_T M \sqrt{D}.$$

Now, note that of the total $p$ periods between synchronizations, only at most $R$ periods will not have event $E$ be true. Therefore, the total regret over all these periods can be bound as,

$$\text{Regret}(T, \bar{E}) \leq R \cdot 2\nu\bar{\beta}_T M \sqrt{D}$$

$$\leq 2\nu\bar{\beta}_T M \sqrt{D} \left( d \log_\nu \left( \frac{\rho_{\max}}{\rho_{\min}} + \frac{TL^2}{d\rho_{\min}} \right) + 1 \right).$$

Adding it all up together gives us,

$$\text{Regret}(T) = \text{Regret}(T, E) + \text{Regret}(T, \bar{E})$$

$$\leq 2\nu\bar{\beta}_T \left[ \sqrt{2MTd \log_\nu \left( \frac{\rho_{\max}}{\rho_{\min}} + \frac{TL^2}{d\rho_{\min}} \right)} + M\sqrt{D} \left( d \log_\nu \left( \frac{\rho_{\max}}{\rho_{\min}} + \frac{TL^2}{d\rho_{\min}} \right) + 1 \right) \right]$$

Setting $D = 2Td \left( \log_\nu \left( \frac{\rho_{\max}}{\rho_{\min}} + \frac{TL^2}{d\rho_{\min}} \right) + 1 \right)^{-1}$, we have:

$$\text{Regret}(T) \leq 4\nu\bar{\beta}_T \sqrt{2MTd \left( \log_\nu \left( \frac{\rho_{\max}}{\rho_{\min}} + \frac{TL^2}{d\rho_{\min}} \right) + 1 \right)}$$

$$\leq 4\nu \left( \sigma\sqrt{2\log\frac{2}{\alpha} + d\log\left( \frac{\rho_{\max}}{\rho_{\min}} + \frac{TL^2}{d\rho_{\min}} \right)} + S\sqrt{M\rho_{\max}} + \kappa\sqrt{M} \right) \sqrt{2MTd \left( \log_\nu \left( \frac{\rho_{\max}}{\rho_{\min}} + \frac{TL^2}{d\rho_{\min}} \right) + 1 \right)}$$

Optimizing over $\nu$ gives us the final result. Asymptotically, we have (setting $\nu = e$), with probability at least $1 - \alpha$,

$$\text{Regret}(T) = O\left( \sigma\sqrt{MTd} \left( \log\left( \frac{\rho_{\max}}{\rho_{\min}} + \frac{TL^2}{d\rho_{\min}} \right) + \sqrt{\log\frac{2}{\alpha}} \right) \right). \tag{10}$$

$\square$

**Theorem 4** (Decentralized Private Group Regret, Theorem 2 of the main paper). *Assuming Proposition 2 holds, and synchronization occurs in at least $n = \Omega\left( d(\bar{\chi}(\mathcal{G}_\gamma) \cdot \gamma)(1 + L^2)^{-1} \log\left( \rho_{\max}/\rho_{\min} + TL^2/d\rho_{\min} \right) \right)$ rounds, decentralized* FEDUCB *obtains the following group pseudoregret with probability at least $1 - \alpha$:*

$$\mathcal{R}_M(T) = O\left( \sigma\sqrt{M(\bar{\chi}(\mathcal{G}_\gamma) \cdot \gamma)Td} \left( \log\left( \frac{\rho_{\max}}{\rho_{\min}} + \frac{TL^2}{\gamma d\rho_{\min}} \right) + \sqrt{\log\frac{2}{\alpha}} + S\sqrt{M\rho_{\max}} + \kappa\sqrt{M} \right) \right).$$

*Proof.* The proof for this setting is similar to the centralized variant. We can first partition the power graph $\mathcal{G}_\gamma$ of the communication network $\mathcal{G}$ into a clique cover $\mathcal{C} = \cup_i C_i$. The overall regret can then be decomposed as the following.

$$\text{Regret}(T) = \sum_{i=1}^{M} \sum_{t=1}^{T} r_{i,t}$$

$$= \sum_{C \in \mathcal{C}} \sum_{i \in C} \sum_{t=1}^{T} r_{i,t}$$

$$= \sum_{C \in \mathcal{C}} \text{Regret}_C(T).$$

Here, $\text{Regret}_C(T)$ denotes the cumulative pseudoregret of all agents within the clique $C$. It is clear that since there is no communication between agents in different cliques, their behavior is independent

---

**Algorithm 3** DECENTRALIZED FEDUCB$(D, M, T, \rho_{\min}, \rho_{\max}, \mathcal{G}, \gamma)$

---

1: **Initialization**: $\forall i, \forall g \in [\gamma]$, set $\boldsymbol{S}_{i,1}^{(g)} \leftarrow \boldsymbol{0}, \boldsymbol{s}_{i,1}^{(g)} \leftarrow \boldsymbol{0}, \boldsymbol{H}_{i,0}^{(g)} \leftarrow \boldsymbol{0}, \boldsymbol{h}_{i,0}^{(g)} \leftarrow \boldsymbol{0}, \boldsymbol{U}_{i,1}^{(g)} \leftarrow \boldsymbol{0}, \bar{\boldsymbol{u}}_{i,1}^{(g)} \leftarrow \boldsymbol{0}$.

2: **for** each iteration $t \in [T]$ **do**

3:     **for** each agent $i \in [M]$ **do**

4:         Set subsampling index $g \leftarrow t \mod \gamma$.

5:         Run lines 4-13 of Algorithm 1 with $\boldsymbol{S}_{i,t}^{(g)}, \boldsymbol{s}_{i,t}^{(g)}, \boldsymbol{U}_{i,t}^{(g)}, \bar{\boldsymbol{u}}_{i,t}^{(g)}, \boldsymbol{V}_{i,t}^{(g)}, \tilde{\boldsymbol{u}}_{i,t}^{(g)}, \boldsymbol{H}_{i,t}^{(g)}, \boldsymbol{h}_{i,t}^{(g)}$

6:         **if** $\log\left( \frac{\det\left(\boldsymbol{V}_{i,t}^{(g)} + \boldsymbol{x}_{i,t}^{(g)}(\boldsymbol{x}_{i,t}^{(g)})^\top + M(\rho_{\max} - \rho_{\min})\boldsymbol{I}\right)}{\det\left(\boldsymbol{S}_{i,t}^{(g)}\right)} \right) \geq \frac{D}{(\Delta t_{i,g}+1)(1+L^2)}$ **then**

7:             REQUEST TO SYNCHRONIZE$(i, g) \leftarrow$ TRUE.

8:         **end if**

9:         **if** REQUEST TO SYNCHRONIZE$(i, g)$ **then**

10:           BROADCAST Message SYNCHRONIZE$(i, g, t)$ from agent $i$ at time $t$ to all neighbors.

11:         **end if**

12:         **for** message $m$ received at time $t$ by agent $i$ **do**

13:             **if** $m = $ SYNCHRONIZE$(i', g', t')$ **then**

14:                **if** $i'$ belongs to the same clique as $i$ and $t' \geq t - \gamma$ **then**

15:                    Agent sends $\widehat{\boldsymbol{Q}}_{i,t}^{(g')} \rightarrow$ PRIVATIZER and gets $\widehat{\boldsymbol{U}}_{i,t+1}^{(g')}, \widehat{\boldsymbol{u}}_{i,t+1}^{(g')} \leftarrow$ PRIVATIZER.

16:                    Agent broadcasts $\widehat{\boldsymbol{U}}_{i,t+1}^{(g')}, \widehat{\boldsymbol{u}}_{i,t+1}^{(g')}$ to all neighbors.

17:                **end if**

18:             **if** $m = \widehat{\boldsymbol{U}}_{i',t'+1}^{(g')}, \widehat{\boldsymbol{u}}_{i',t'+1}^{(g')}$ **then**

19:                **if** $i'$ belongs to the same clique as $i$ and $t' \geq t - \gamma$ **then**

20:                    Agent updates $\boldsymbol{S}_{i,t+1}^{(g')}, \boldsymbol{S}_{i,t+1}^{(g')}$ with $\widehat{\boldsymbol{U}}_{i',t'+1}^{(g')}, \widehat{\boldsymbol{u}}_{i',t'+1}^{(g')}$.

21:                **end if**

22:             **end if**

23:             **end if**

24:         **end for**

25:     **end for**

26: **end for**

---

and we can analyse each clique separately. Now, consider $\kappa$ sequences given by $s_1, ..., s_\kappa$, where $s_i = (i, i+\kappa, i+2\kappa, ..., i + (\lceil T/\kappa \rceil - 1)\kappa)$. For any clique $C$ we can furthermore decompose the regret as follows.

$$
\begin{aligned}
\text{Regret}_C(T) &= \sum_{i \in C} \sum_{t=1}^{T} r_{i,t} \\
&= \sum_{i \in C} \sum_{j=1}^{\kappa} \sum_{t \in s_j} r_{i,t} \\
&= \sum_{j=1}^{\gamma} \text{Regret}_{C,j}(T).
\end{aligned}
$$

Here $\text{Regret}_{C,j}(T)$ denotes the cumulative pseudoregret of the $j^{th}$ subsequence. We will now bound each of these regret terms individually, with an identical argument as the centralized case. This can be done since the behavior of the algorithm in each of these subsequences is independent: each sequence $s_j$ corresponds to a different Gram matrix and least-squares estimate, and is equivalent to each agent running $\gamma$ parallel bandit algorithms. We now bound each of the regret terms $\text{Regret}_{C,j}(T)$ via an identical argument as the centralized case. Let $\bar{\boldsymbol{\sigma}}_j$ be the subsequence of $[T]$ containing every $j^{th}$ index, i.e., $\bar{\boldsymbol{\sigma}}_j = j, j + \gamma, j + 2\gamma, ..., j + \lceil T/\gamma - 1 \rceil \gamma$. For any clique $C$, and index $j$ we compare the pulls of each agent within $C$ to the pulls taken by an agent pulling arms in a round-robin manner $(\boldsymbol{x}_{i,j})_{i \in C, j \in \bar{\boldsymbol{\sigma}}_j}$. This corresponds to a total of $|C|T/\gamma$ pulls.

Now, it is crucial to see that according to Algorithm 3, if a signal to synchronize has been sent by any agent $i \in C$ at any time $t$ (belonging to subsequence $j$), then the $j^{th}$ parameter set $\boldsymbol{V}_{i,t}^{(j)}$ is used first at time $t$ (by each agent in $C$), then at time $t + \gamma$ (at which time each agent in $C$ broadcasts their parameters, since by this time each other agent will have received the signal to synchronize, as they are at most distance $\gamma$ apart), and next at time $t + 2\gamma$, upon which they will be fully synchronized.

Now, if we denote the rounds $n_{C,j} \subset \bar{\boldsymbol{\sigma}}_j$ as the rounds in which each agent broadcasted their $j^{th}$ parameter sets, then for any $\tau \in n_{C,j}$, all agents in $C$ have identical $j^{th}$ parameter sets at instant $\tau + 1$, and for each instant $\tau - 1$, all agents in $C$ will obey the synchronization threshold for the $j^{th}$ set of parameters, (log-det condition).

Now, we denote by $\boldsymbol{W}_{i,t}^{(j,C)}$ the Gram matrix obtained by the hypothetical round-robin agent for subsequence $j$ in clique $C$. By Lemma 2, we have that

$$\sum_{t \in \bar{\boldsymbol{\sigma}}_j} \sum_{i \in C} \|\boldsymbol{x}_{i,t}\|^2_{(\boldsymbol{W}_{i,t}^{(j,C)})^{-1}} \leq 2d \log\left(1 + \frac{TL^2}{\gamma d \rho_{\min}}\right). \tag{11}$$

After any round $T_k$ of synchronization, consider the cumulative Gram matrices of all observations obtained until that round as $\boldsymbol{V}_k^{(j,C)}$, $k = 1, ..., n_{j,C} - 1$, regularized by $|C|\rho_{\min}\boldsymbol{I}$, i.e., $\boldsymbol{V}_k^{(j,C)} = \sum_{i \in C} \sum_{t \in \bar{\boldsymbol{\sigma}}_j : t < T_k} \boldsymbol{x}_{i,t}\boldsymbol{x}_{i,t}^\top + |C|\rho_{\min}\boldsymbol{I}$. Finally, let $\boldsymbol{V}_p$ denote the (regularized) $j^{th}$ Gram matrix with all trials from agents within $C$ at time $T$, and $\boldsymbol{V}_0^{(j,C)} = |C|\rho_{\min}\boldsymbol{I}$. Therefore, we have that $\det(\boldsymbol{V}_0^{(j,C)}) = (|C|\rho_{\min})^d$, and that $\det(\boldsymbol{V}_{n_{j,C}}) \leq \left(\frac{\text{tr}(\boldsymbol{V}_{n_{j,C}}^{(j,C)})}{d}\right)^d \leq (|C|\rho_{\max} + |C|TL^2/(\gamma d))^d$. Therefore, for any $\nu > 1$,

$$\log_\nu\left(\frac{\det(\boldsymbol{V}_{n_{j,C}}^{(j,C)})}{\det(\boldsymbol{V}_0^{(j,C)})}\right) \leq d \log_\nu\left(\frac{\rho_{\max}}{\rho_{\min}} + \frac{TL^2}{\gamma d \rho_{\min}}\right).$$

Let $R = \left\lceil d \log_\nu\left(\frac{\rho_{\max}}{\rho_{\min}} + \frac{TL^2}{\gamma d \rho_{\min}}\right)\right\rceil$. It follows that in all but $R$ periods between synchronization,

$$1 \leq \frac{\det(\boldsymbol{V}_k^{(j,C)})}{\det(\boldsymbol{V}_{k-1}^{(j,C)})} \leq \nu. \tag{12}$$

We consider the event $E$ to be the period $k$ when Equation 12 holds. Now, for any $T_{k-1} \leq t \leq T_k$, consider the immediate pseudoregret for any agent $i$. By Lemma 1, we have for any agent $i \in C$ and $t \in \bar{\boldsymbol{\sigma}}_j$:

$$r_{i,t} \leq 2\bar{\beta}_T \|x_{i,t}\|_{(\boldsymbol{V}_{i,t}^{(j,C)})^{-1}}$$

$$\leq 2\bar{\beta}_T \|x_{i,t}\|_{(\boldsymbol{G}_{i,t}^{(j,C)} + |C|\rho_{\min}\boldsymbol{I})^{-1}} \qquad (\boldsymbol{V}_{i,t}^{(j,C)} \succcurlyeq \boldsymbol{G}_{i,t}^{(j,C)} + |C|\rho_{\min}\boldsymbol{I})$$

$$\leq 2\bar{\beta}_T \|x_{i,t}\|_{(\boldsymbol{W}_{i,t}^{(j,C)})^{-1}} \cdot \sqrt{\frac{\det(\boldsymbol{W}_{i,t}^{(j,C)})}{\det(\boldsymbol{G}_{i,t}^{(j,C)} + |C|\rho_{\min}\boldsymbol{I})}}$$

$$\leq 2\bar{\beta}_T \|x_{i,t}\|_{(\boldsymbol{W}_{i,t}^{(j,C)})^{-1}} \cdot \sqrt{\frac{\det(\boldsymbol{V}_k^{(j,C)})}{\det(\boldsymbol{G}_{i,t}^{(j,C)} + |C|\rho_{\min}\boldsymbol{I})}} \qquad (\boldsymbol{V}_k^{(j,C)} \succcurlyeq \boldsymbol{W}_{i,t}^{(j,C)})$$

$$\leq 2\bar{\beta}_T \|x_{i,t}\|_{(\boldsymbol{W}_{i,t}^{(j,C)})^{-1}} \cdot \sqrt{\frac{\det(\boldsymbol{V}_k^{(j,C)})}{\det(\boldsymbol{V}_{k-1}^{(j,C)})}} \qquad (\boldsymbol{G}_{i,t}^{(j,C)} + |C|\rho_{\min}\boldsymbol{I} \succcurlyeq \boldsymbol{V}_{k-1}^{(j,C)})$$

$$\leq 2\nu\bar{\beta}_T \|x_{i,t}\|_{(\boldsymbol{W}_{i,t}^{(j,C)})^{-1}}. \qquad \text{(Event } E \text{ holds)}$$

Now, we can sum up the immediate pseudoregret over all such periods where $E$ holds to obtain the total regret for these periods. With probability at least $1 - \alpha$,

$$\text{Regret}_{C,j}(T, E) = \sum_{i \in C} \sum_{t \in \bar{\boldsymbol{\sigma}}_j : \text{E is true}} r_{i,t}$$

$$\leq \sqrt{\frac{|C|T}{\gamma} \left( \sum_{i \in C} \sum_{t \in \bar{\boldsymbol{\sigma}}_j : \text{E is true}} r_{i,t}^2 \right)}$$

$$\leq 2\nu\bar{\beta}_T \sqrt{\frac{|C|T}{\gamma} \left( \sum_{i \in C} \sum_{t \in \bar{\boldsymbol{\sigma}}_j : \text{E is true}} \|\boldsymbol{x}_{i,t}\|_{(\boldsymbol{W}_{i,t}^{(j,C)})^{-1}} \right)}$$

$$\leq 2\nu\bar{\beta}_T \sqrt{\frac{|C|T}{\gamma} \left( \sum_{i \in C} \sum_{t \in \bar{\boldsymbol{\sigma}}_j} \|\boldsymbol{x}_{i,t}\|_{(\boldsymbol{W}_{i,t}^{(j,C)})^{-1}} \right)}$$

$$\leq 2\nu\bar{\beta}_T \sqrt{2\frac{|C|T}{\gamma} d \log_\nu \left( 1 + \frac{TL^2}{\gamma d \rho_{\min}} \right)}.$$

Now let us consider the periods in which $E$ does not hold for any subsample $j$. In any such period between synchronization of length $t_k = T_k - T_{k-1}$, we have, for any agent $i$, the regret accumulated given by:

$$\text{Regret}_{C,j}([T_{k-1}, T_k]) = \sum_{t=T_{k-1}}^{T_k} \sum_{i \in C} r_{i,t}$$

$$\leq 2\nu\bar{\beta}_T \left( \sum_{i \in C} \sqrt{t_k \sum_{t=T_{k-1}}^{T_k} \|\boldsymbol{x}_{i,t}\|_{(\boldsymbol{V}_{i,t}^{(j)})^{-1}}^2} \right)$$

$$\leq 2\nu\bar{\beta}_T \left( \sum_{i \in C} \sqrt{t_k \log_\nu \left( \frac{\det(\boldsymbol{V}_{i,t+t_k}^j)}{\det(\boldsymbol{V}_{i,t}^j)} \right)} \right)$$

$$\leq 2\nu\bar{\beta}_T \left( \sum_{i \in C} \sqrt{t_k \log_\nu \left( \frac{\det(\boldsymbol{G}_{i,t+t_k}^j + |C|\rho_{\max}\boldsymbol{I})}{\det(\boldsymbol{G}_{i,t}^j + |C|\rho_{\min}\boldsymbol{I})} \right)} \right)$$

By Algorithm 1, we know that for all agents, $t_k \log_\nu \left( \frac{\det(\boldsymbol{G}_{i,t+t_k} + M\rho_{\max}\boldsymbol{I})}{\det(\boldsymbol{G}_{i,t} + M\rho_{\min}\boldsymbol{I})} \right) \leq D$ (since there would be a synchronization round otherwise), therefore

$$\leq 2\nu\bar{\beta}_T |C| \sqrt{D}.$$

Now, note that of the total $p$ periods between synchronizations, only at most $R$ periods will not have event $E$ be true. Therefore, the total regret over all these periods can be bound as,

$$\text{Regret}_{C,j}(T, \bar{E}) \leq R \cdot 2\nu\bar{\beta}_T |C| \sqrt{D}$$

$$\leq 2\nu\bar{\beta}_T |C| \sqrt{D} \left( d \log_\nu \left( \frac{\rho_{\max}}{\rho_{\min}} + \frac{TL^2}{d\rho_{\min}} \right) + 1 \right).$$

In a manner identical to the centralized case, we can obtain the total pseudoregret within a clique $C$ for subsampling index $j$ by choosing an appropriate value of $D$. Note that since the broadcast between agents happens at an additional delay of 1 trial, the value $D$ must be scaled by a factor of $1 + \sup_{\boldsymbol{x} \in \mathcal{D}_{i,t}} \|\boldsymbol{x}\|_2^2$ to bound the additional term in the determinant (by the matrix-determinant lemma), giving us the extra $1 + L^2$ term from the proof. Finally, summing up the above over all $C \in \mathcal{C}$ and $j \in [\gamma]$ and noting that $|C| \leq M \forall C \in \mathcal{C}$, and that $|\mathcal{C}| = \bar{\chi}(\mathcal{G}_\gamma)$ gives us the final result. $\square$

Figure 2: An experimental comparison of centralized FEDUCB on varying the minimum gap between arms $\Delta$, for various values of the privacy budget $\rho_{\min}$.

**Proposition 7.** *Fix $\alpha > 0$. If each agent $i$ samples noise parameters $\boldsymbol{H}_{i,t}$ and $\boldsymbol{h}_{i,t}$ using the tree-based Gaussian mechanism mentioned above for all $n$ trials of $\bar{\boldsymbol{\sigma}}$ in which communication occurs, then the following $\rho_{\min}, \rho_{\max}$ and $\kappa$ are $(\alpha/2nM, \bar{\boldsymbol{\sigma}})$-accurate bounds:*

$$\rho_{\min} = \Lambda, \ \rho_{\max} = 3\Lambda, \ \kappa \leq \sqrt{m(L^2 + 1)\left(\sqrt{d} + 2\log(2nM/\alpha)\right)} / (\sqrt{2}\varepsilon).$$

*Proof.* The proof follows directly from Shariff and Sheffet [41], Proposition 11. Instead of $\alpha/2n$, we replace the parameter as $\alpha/2nM$. $\qquad\square$

**Remark 6** (Decentralized Protocol). *Decentralized FEDUCB obtains similar bounds for $\rho_{\min}, \rho_{\max}$ and $\kappa$, with $m = 1 + \lceil\log(T/\gamma)\rceil$. An additional term of $\log(\gamma)$ appears in $\Lambda$ and $\kappa$, since we need to now maintain $\gamma$ partial sums with at most $T/\gamma$ elements in the worst-case (see Appendix). Unsurprisingly, there is no dependence on the network $\mathcal{G}$, as privatization is done at the source itself.*

**Discussion**. In the decentralized version of the algorithm, each agent maintains $\gamma$ sets of parameters that are used in a round-robin manner. Note that this implies that each parameter set is used at most $T/\gamma$ times per agent. This implies that in the worst case, each agent will communicate at most $T/\gamma$ partial sums related to this parameter set, hence needing at most $1 + \lceil\log(T/\gamma)\rceil$ separate nodes of the tree-based mechanism for the particular set of parameters, which leads to the additional $\log\gamma$ in the bounds as well. Next, when determining $\kappa$, each agent will not require $M$-accurate bounds, since it communicates with only $|C|$ other agents. However, in the worst case, a centrally-positioned node belongs to a small clique but can still communicate with all other $M - 1$ nodes (i.e., they can still obtain the partial sums broadcasted), and hence we maintain the factor $M$ to ensure privacy.

## Additional Experiments

In addition to the experiments in the main paper, we conduct ablations with variations in $\Delta$. Figure 2 summarizes the results when run on $M = 10$ agents for different privacy budgets and arm gaps. As expected, the overall regret decreases as the gap increases, and the algorithm becomes less sensitive to privacy budget altogether.

## Footnotes

[1]Originally, federated learning referred to the algorithm proposed in [28] for supervised learning, however, now the term broadly refers to the distributed cooperative learning setting [27].

[2]The *pseudoregret* is an expectation (over the randomness of $\eta_{i,t}$) of the stochastic quantity *regret*, and is more amenable to high-probability bounds. However, a bound over the pseudoregret can also bound the regret with high probability, e.g., by a Hoeffding concentration (see, e.g., [47]).

[3]Assuming bounded rewards is required for the privacy mechanism, and is standard in the literature [3, 2].

[4]Under the stronger assumption that each agent interacts with a completely different set of individuals, we do not need to invoke the composition theorem (as $\boldsymbol{x}_{1,t}, \boldsymbol{x}_{2,t}, ..., \boldsymbol{x}_{M,t}$ are independent for each $t$). However, in the case that one individual could potentially interact simultaneously with all agents, this is not true (e.g., when for some $t$, $\mathcal{D}_{i,t} = \mathcal{D}_{j,t} \; \forall i, j$) and we must invoke the $k$-fold composition Theorem[17] to ensure privacy.