[Reviews · NeurIPS 2020]

Review 1

Summary and Contributions: This paper presents an algorithm and corresponding upper bounds for federated contextual bandits under a differential privacy constraint. They consider both a centralised and distributed setting. While the results are not surprising and the proof methods are relatively standard, it is the first theoretical study of this problem, which has significant practical importance.

Strengths: Rigorous, good theory, with an analysis of both centralised and distributed setting including: - Regret bounds - Communication complexity

Weaknesses: It would be good if there were also some lower bounds for the federated/distributed case.

Correctness: I have not checked thoroughly for correctness but everything appears rigorous

Clarity: The paper is quite clear. However, some details of the model are unclear.

Relation to Prior Work: All directly related work is cited and discussed.

Reproducibility: Yes

Additional Feedback: It is slightly unclear from your DP model whether the agents simply broadcast randomised versions of their actual actions to the mechanism, or if they actually have to take the same actions as the ones they broadcast. It seems to be the latter, as far as I can tell. In l. 36 you say that your regret bound is $1/\sqrt(\epsilon)$ and you say that is a factor $1/\sqrt(\epsilon)$ away from [42]. But I guess that one is $1/\epsilon$, so your bound is lower than the lower bound. This is perhaps not very surprising, as my reading of [42] suggests that the lower bound does not apply to DP bandit algorithims in gneral, but only those that use local differential privacy. Privatizer, however, only adds noise to the actions. However, their reasoning is a bit unclear. Lower bounds would be good, I guess! The decentralized setting is particularly interesting. It uses the notion of delayed feedback to do the analysis, something that has been recently a focus in the regret literature. update : thank you for the clarification


Review 2

Summary and Contributions: The paper describes a distributed learning protocol which balances quality of decisions, communication overhead, and leakage of information. Algorithmic components from the contextual bandit and differential privacy literature are combined and analyzed. ------------------- Having read the other reviews, author's response, and reviewer discussion: I do not see any reason to adjust my overall score.

Strengths: The topic is important and the paper is well written. This particular combination has high potential practical impact and the analysis of the combination is thorough.

Weaknesses: No complaints, solid work.

Correctness: I did not analyze proofs in detail.

Clarity: Very well written.

Relation to Prior Work: Yes.

Reproducibility: No

Additional Feedback: I checked the appendix but did not find additional detailed information about the experiments.


Review 3

Summary and Contributions: The paper considers the problem of federated/distributed contextual linear bandits under differential privacy constraints. More concretely, the goal is to compute the optimal parameters vector  \theta that defines a linear reward function to minimize the regret over a time horizon T. The key contribution claimed by the authors is the first differentially private federated algorithm for this problem. 

Strengths: Contextual bandit is a classic problem and it is indeed, theoretically, interesting to study it from a differential privacy point of view. It is indeed relevant to the theoretical NeurIPS community. 

Weaknesses: I have few key concerns that would be good to get some clarification on. 1) What is the privacy angle? Is there an application where sharing the reward/current parameters would lead to privacy leakage (typically the angle would be either data sensitivity or its proprietary nature, -- what is it here )? 2) Some of the theoretical aspects were very hard to follow. Partly this is because, the paper draws heavily from prior art -- which I am unfamiliar with. On the other hand, I blame it partly on the writeup.   The critical one, which I would really need some clarity on, is the relationship between (epsilon, deltas) and the privacy budget parameters (rho_max, rho_min, and kappa). On page 5, it is mentioned: "In turn, the quantities ε and δ affect the algorithm (and regret) via the quantities ρmin, ρmax, and κ which bound the norm of perturbations at any round:" But these privacy budget quantities ρmin, ρmax, and κ are never specified in terms of epsilon, deltas. Definition 2 is perhaps defining the former quantities in terms of perturbations (H) norms but I do not see their relationship with epsilon, delta. This is critical to understand because, the experiments are performed with respect to privacy budget parameters (\rhos and kappa). It is important to understand the plots in terms of the standard DP parameters (namely epsilon and delta).

Correctness: Methodology is sound but I would like clarity on the privacy budgets as mentioned above. I have not verified the proofs. 

Clarity: For a non-expert the paper is really hard to read/follow. The presentation should be (if accepted) made better.

Relation to Prior Work: Somewhat, but I am not sure how distinguished the techniques are from the prior approaches. While I appreciate that the challenges primarily stem from the asymmetric communication between agents, I cannot judge on how complicated this really is and what new techniques are employed here. 

Reproducibility: Yes

Additional Feedback: Thank you authors for the response to my questions. Based on that and discussions with fellow reviewers and Area chair, I have decided to increase my score. However, for the sake of better understandability, please address the concerns I have raised before mainly that the privacy analysis must be augmented with epsilon and delta values used.


Review 4

Summary and Contributions: This work introduces FedUCB - an algorithm for differentially-privates contextual bandits (CB) in a centralized and decentralized federated learning setting, with the main contributions of the paper being: 1. Defining differential privacy for federated CBs 2. Introducing a centralized federated learning algorithm for CB with regret bounds that match single agent regret when synchronization happens every round, and 3. Introducing a decentralized version of the algorithm, with regret bounds that additionally depend on the topology of the graph.

Strengths: The problem that the work targets is relevant and well-motivated, and the work is technically sound, novel, and thorough in its presentation.

Weaknesses: would like to see discussion on non-homogeneous arms for different agents and the effect of new arms being introduced in the course of an experiment. Have the authors considered if we can have a privacy budget that is dependent on each pair of agents? Meaning some agents are allowed to share more compared to others?

Correctness: Although have not check all the proofs, parts that I needed to reproduce to get deeper understanding seems all to be correct.

Clarity: The paper is clearly written and expands the problem and its motivation in a nice way.

Relation to Prior Work: The introduction is easy to read and follow and covers mostly what has been done in the field. I didn't search extensively for any other prior work that needs to be added though.

Reproducibility: Yes

Additional Feedback: page 8, Line 276 . "Beta" should be /beta .

[Author Response · NeurIPS 2020]

# Paper ID 974 Rebuttal

We sincerely thank all reviewers for their time and thoughtful feedback. We address concerns sequentially.

**Reviewer 1.**

**1. Actions being broadcast**: Apologies for the confusion! Indeed, the actions (and rewards) broadcast are different from those taken. For example, if communication occurs every round, then the agent broadcasts a perturbed Gram matrix of *all rewards* up to that instant, (i.e., even at $t = 1$, the rank-1 matrix broadcasted is additionally perturbed as well). The original actions or rewards are not transmitted anywhere. We will be sure to clarify this in the final version.

**2. Lower Bound.** Thanks for the catch! We overlook a detailed comparison of the lower bound in the draft, which is crucial since the comparison is not straightforward, thanks again for catching this. The bound presented in Shariff and Sheffet (2018) is for the case when the arm rewards are separated by a gap $\Delta$ (and hence the $\mathcal{O}(\log T/\varepsilon)$ bound). Our $\varepsilon$-dependent bound in the same case admits a dependence of $\mathcal{O}((\log T/\varepsilon)^{3/2})$, which is an excess of $1/\sqrt{\varepsilon}$. We will definitely address lower bounds in more detail in the full paper, apologies for the confusion!

**Reviewer 2.**

Thank you for your review and positive appraisal of the paper! Apologies for the detailed experiment information – essentially all experiments utilize the identical setting of Section 4.4, and should be reproducible from Section 4.4. (up to randomness of the environment). The algorithms have been implemented in Python using NumPy, following the library `contextualbandits` as reference.

**Reviewer 3.**

**1. Privacy Angle.** Thank you for the question! We apologize for the confusion. Contextual bandit algorithms are most relevant in recommendation systems, where the context vectors $x_t$ usually refer to a (random) user's description at time $t$, which often includes sensitive information about the user (e.g., in online retail, it will include a vector of websites visited, etc.), and hence this information is desired to be kept private. Moreover, in our setting, several agencies cooperate to solve the problem (in the decentralized setting), which requires privacy mechanisms to be set in place. For example, in medical imaging, a group of hospitals may be interested in training a joint model, however, none wish to share their data as per regulations. Our approach can enable joint learning in this setting.

**Privacy parameters.** Apologies for the unclear exposition! Approximate differential privacy assumes a noise threshold ($\varepsilon$) and a failure probability ($\delta$); both these parameters are fixed during the design of the algorithm. Now, our algorithm builds on the idea of changing regularizers, and has 3 crucial parameters $\rho_{\min}, \rho_{\max}$ and $\kappa$. Proposition 4 in the paper describes how, for any $(\varepsilon, \delta)$ one can obtain the required values of $\rho_{\min}$ and $\rho_{\max}$ (since $\kappa$ only shows up in the regret, and not the algorithm itself). These quantities, in turn, provide a bound on the group regret as per Theorem 1. Simply replacing these quantities in terms of $\varepsilon$ and $\delta$ gives us a regret bound in terms of the privacy parameters themselves, as in Corollary 1.

In a nutshell, in the experiments, we vary $\rho_{\min}$ directly (following the protocol in Shariff and Sheffet (2018)), but we will include a comparison with the privacy parameter $\varepsilon$ directly, as that will definitely improve understanding of the algorithm. Thank you for this question! we will also include a brief comment on the privacy parameters to clarify the setting.

**Reviewer 4.**

Thank you for your positive appraisal and catching the typo!

[Meta-Review · NeurIPS 2020]

Three of the four reviewers evaluated this paper very positively, noting the good, rigorous theory and the relevance of the topic of federated bandits with privacy guarantees, and judged the work as potentially high-impact. The author rebuttal was taken into account and the subsequent discussion led the 4th (less positive) reviewer to increase his/her score. Therefore, this paper is accepted. However, for the final version, the authors are asked to explicitly report the values of epsilon and delta in the experiments, as standard in the differential privacy literature.